# Functional Matching of Logic Subgraphs: Beyond Structural Isomorphism

**Ziyang Zheng   Kezhi Li   Zhengyuan Shi   Qiang Xu**
The Chinese University of Hong Kong
`{zyzheng23,kzli24,zyzshi21,qxu}@cse.cuhk.edu.hk`

## Abstract

Subgraph matching in logic circuits is foundational for numerous Electronic Design Automation (EDA) applications, including datapath optimization, arithmetic verification, and hardware trojan detection. However, existing techniques rely primarily on structural graph isomorphism and thus fail to identify function-related subgraphs when synthesis transformations substantially alter circuit topology. To overcome this critical limitation, we introduce the concept of *functional subgraph matching*, a novel approach that identifies whether a given logic function is implicitly present within a larger circuit, irrespective of structural variations induced by synthesis or technology mapping. Specifically, we propose a two-stage multi-modal framework: (1) learning robust functional embeddings across AIG and post-mapping netlists for functional subgraph detection, and (2) identifying fuzzy boundaries using a graph segmentation approach. Evaluations on standard benchmarks (ITC99, OpenABCD, ForgeEDA) demonstrate significant performance improvements over existing structural methods, with average $93.8\%$ accuracy in functional subgraph detection and a dice score of $91.3\%$ in fuzzy boundary identification. The source code and implementation details can be found at *our repository*.

## 1   Introduction

Subgraph matching—identifying smaller graphs within larger ones—is a fundamental task in graph analysis, with pivotal applications spanning social network mining, bioinformatics, and Electronic Design Automation (EDA).

In the context of EDA, subgraph matching involves searching for specific circuit patterns embedded within larger circuits. This capability directly supports critical tasks such as circuit optimization, verification, and security analyses. For example, verifying complex arithmetic circuits like multipliers typically requires recognizing embedded small functional units (e.g., half-adders) within larger netlists, enabling algebraic simplifications and correctness proofs [1, 2]. Similarly, during template-based synthesis, accurately locating predefined subgraphs allows their replacement with highly optimized standard cells, thereby significantly improving power, performance, and area (PPA) metrics [3]. Moreover, subgraph matching also plays an essential role in hardware security by enabling the identification of potentially malicious substructures or "hardware trojans"—anomalous subcircuits intentionally embedded to compromise system integrity [4, 5].

Traditionally, subgraph matching in graphs is formulated as a *structural isomorphism* problem: determining whether a smaller query graph exactly matches part of a larger target graph in terms of node and edge connectivity. This problem is extensively studied in general graph theory, and classical approaches rely primarily on combinatorial search algorithms [6, 7, 8]. However, subgraph isomorphism is an NP-complete problem and thus often suffers from exponential computational complexity in worst-case scenarios. Recently, deep learning methods have emerged to mitigate this computational cost by embedding graphs into continuous latent spaces, significantly accelerating

39th Conference on Neural Information Processing Systems (NeurIPS 2025).

matching tasks [9, 10, 11]. Within the EDA domain, these techniques have been successfully adapted for transistor-level subcircuit identification [12].

However, structure-based matching methods encounter significant limitations in practical EDA tasks, as circuit topologies frequently undergo substantial transformations during logic synthesis and technology mapping. Equivalent logic functions can thus be realized through widely differing structural implementations, driven by design considerations such as timing performance, power consumption, or silicon area. Consequently, exact structural correspondence rarely persists throughout the design process, even when the underlying logic function remains unchanged. This inherent limitation severely restricts the utility of traditional structural matching techniques, particularly in applications requiring cross-stage queries—for example, identifying subgraphs from an abstract netlist (like an And-Inverter Graph, or AIG) within a synthesized, technology-mapped netlist.

Motivated by this critical gap, we introduce an approach explicitly designed to recognize logic functionality irrespective of structural differences. Specifically, our framework determines whether the logic represented by a query subgraph exists implicitly within a candidate graph, independent of structural transformations.

To formalize this, we propose two key concepts: (1) **functional subgraph**, representing the circuit logic containment relation independent of structure, and (2) **fuzzy boundary**, minimal graph regions encapsulating the query's logic despite unclear structural boundaries. Consequently, our methodology, termed *functional subgraph matching*, addresses two sub-tasks: *1. Functional Subgraph Detection*: Determining whether the logic function of a query graph is implicitly contained within a candidate graph; *2. Fuzzy Boundary Identification*: Precisely locating the smallest possible region (the fuzzy boundary) in the candidate graph that encapsulates the query's logic.

To achieve these objectives, we propose a novel two-stage multi-modal framework. In the first stage, we train our model with intra-modal and inter-modal alignment across different graph modalities, enabling robust and cross-stage detection of functional subgraph. In the second stage, we fine-tune our model and formulate fuzzy boundary detection as a graph segmentation task, moving beyond prior approaches that treated boundary identification as an input-output classification problem [13, 14]. By leveraging information from nodes located within the true boundaries, our segmentation approach significantly enhances performance and continuity of fuzzy boundary prediction.

Our experiments demonstrate the effectiveness of the proposed framework. Evaluations conducted across several widely-used benchmarks, ITC99 [15], OpenABCD [16] and ForgeEDA [17], show that our approach significantly surpasses traditional structure-based methods. Specifically, our framework achieves an average accuracy of 93.8% for functional subgraph detection and attains a DICE score of 91.3% for fuzzy boundary detection tasks. In contrast, structure-based baseline methods typically exhibit near-random performance (accuracy close to 50%) and high variability in precision, recall, and F1-score, underscoring their limitations in capturing implicit functionality. To further validate our method's robustness and generalizability, we additionally propose three function-aware baseline variants by integrating different graph encoders into our framework.

In summary, the contributions of this work include:

- Introducing and formally defining the novel concept of functional subgraph matching, clearly distinguishing it from structural isomorphism and functional equivalence.

- Developing a two-stage multi-modal embedding framework, leveraging both intra-modal and inter-modal alignments to capture structure-agnostic and function-invariant graph representations. This allows effective functional subgraph detection across different modalities.

- Proposing an innovative approach for fuzzy boundary identification by formulating the task as a graph segmentation problem rather than a simple input-output classification problem, significantly enhancing boundary continuity and localization accuracy.

## 2 Preliminaries

### 2.1 Subgraph Isomorphism Matching

Subgraph isomorphism matching is a fundamental problem in graph theory with applications across bioinformatics [18], social network analysis [19], and knowledge graphs [20, 21]. We first recall the standard definition of subgraph isomorphism in Definition 1.

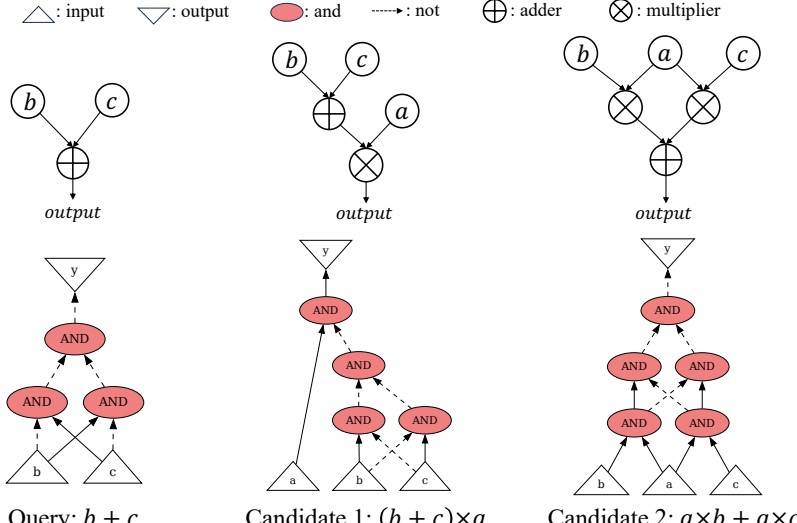

Figure 1: We present the query and candidate graphs. **Above**: 1-bit adder and multiplier. **Below**: AIG netlist. The query $b + c$ is explicitly contained within the candidate $(b + c) \times a$, making it straightforward to identify the exact subgraph in the candidate. In contrast, the query $b + c$ is implicitly contained within the candidate $a \times b + a \times c$, which implies no subgraph of $a \times b + a \times c$ has the same structure or function as the query graph.

**Definition 1** (**Subgraph Isomorphism**). *A graph $\mathcal{Q}$ is an **isomorphic subgraph** of $\mathcal{G}$ if there exists a subgraph $\mathcal{G}'$ of $\mathcal{G}$ such that $\mathcal{Q}$ is isomorphic to $\mathcal{G}'$.*

Then, based on the definition of subgraph isomorphism, the subgraph isomorphism matching task is defined as follows: given a query graph $\mathcal{Q}$ and a target graph $\mathcal{G}$, determine if $\mathcal{Q}$ is isomorphic to a subgraph of $\mathcal{G}$. Classical approaches of subgraph isomorphism matching rely primarily on combinatorial search algorithms [7, 8, 6]. Its NP-complete nature, however, makes exact matching computationally intensive. More recently, graph-neural-network-based (GNN-based) methods have been introduced to learn compact graph embeddings that accelerate the matching process [9, 10, 11]. In the EDA domain, Li et al. [12] adapt the NeuroMatch architecture [10] to solve subcircuit isomorphism on transistor-level netlists.

However, in EDA flow, graphs often represent circuits or computations where structural modifications can preserve the underlying function. Standard subgraph isomorphism struggles with such cases. For instance, as illustrated in Figure 1, a model based on Definition 1 can identify that the structure representing $b + c$ is contained within $a \times (b + c)$, but it cannot identify the functional presence of $b + c$ within the structurally different but functionally related expression $a \times b + a \times c$.

## 2.2 Subgraph Equivalence

The limitation of structure-based subgraph matching motivates considering functional properties. Function-aware representation learning has emerged as a pivotal subfield in EDA. Many recent works emphasize functional equivalence, denoted $\mathcal{G}_1 \equiv_{func} \mathcal{G}_2$. DeepGate [22, 23, 24] and DeepCell [25] employ disentanglement to produce separate embeddings for functionality and structure, pretraining across various EDA benchmarks and predict functional similarity with a task head. PolarGate [26] enhances functional embeddings by integrating ambipolar device principles. FGNN [27, 28] applies contrastive learning to align circuit embeddings according to functional similarity.

While graph isomorphism requires structural identity, functional equivalence relates graphs based on their input-output behavior. Building on this, we can define a notion of subgraph relationship based on function, as shown in Definition 2.

**Definition 2** (**Subgraph Equivalence**). *A graph $\mathcal{Q}$ is an **equivalent subgraph** of $\mathcal{G}$ if there exists a subgraph $\mathcal{G}'$ of $\mathcal{G}$ such that $\mathcal{Q} \equiv_{func} \mathcal{G}'$.*

This definition allows for functional matching within existing subgraphs. Some works adopt similar ideas for tasks such as arithmetic block identification [13, 29] and symbolic reasoning [14, 30], which aim to find a subgraph with specific functionality rather than structure. Compared to subgraph isomorphism, subgraph equivalence offers more flexibility against local structure modifications. However, Definition 2 still falls short for cases involving global restructuring. As shown in Figure 1, in the example $a \times b + a \times c$, no single subgraph is functionally equivalent to $b + c$. The function $b + c$ is **implicitly** present but not explicitly represented by a contiguous subgraph.

## 2.3 Functional Subgraph

To address the limitations of both Definition 1 and Definition 2, we introduce the concept of a functional subgraph, which aims to identify the implicit containment relation between graphs.

**Definition 3** (**Functional Subgraph**). *A graph $\mathcal{Q}$ is a **functional subgraph** of $\mathcal{G}$, denoted $\mathcal{Q} \preccurlyeq \mathcal{G}$, if there exists a graph $\mathcal{G}'$ such that $\mathcal{G}' \equiv_{func} \mathcal{G}$ and $\mathcal{Q}$ is isomorphic to a subgraph of $\mathcal{G}'$.*

This definition captures the idea that the query's function is implicitly contained within the target's function, even if the target's structure has undergone functional transformations, and no exact subgraph isomorphic to the query graph can be found in the target graph. By this definition, we know that $b + c$ is a functional subgraph of $a \times b + a \times c$ since $a \times b + a \times c \equiv_{func} a \times (b + c)$ and $b + c$ is an isomorphic subgraph of $a \times (b + c)$. Furthermore, Definition 3 encompasses Definition 2, i.e., Definition 2 is a special case of Definition 3, as discussed in Proposition 1 (proof in Appendix A).

**Proposition 1.** *If a graph $\mathcal{Q}$ is an equivalent subgraph of $\mathcal{G}$, then $\mathcal{Q}$ is a functional subgraph of $\mathcal{G}$.*

**Properties of Functional Subgraph**   In this paper, we assume that a graph obtained by removing some nodes and edges is not functionally equivalent to the original graph, i.e. $\forall g \neq \emptyset, G \setminus g \not\equiv_{\text{func}} G$. For example, we consider it illegal to directly connect two NOT gates. Therefore, such connections do not appear in our graph structures. In fact, EDA tools such as ABC [31] inherently enforce this constraint. According to Definition 3, functional subgraphs exhibit the following properties:

- **Reflexivity**: For any graph $\mathcal{G}$, $\mathcal{G}$ is the functional subgraph of $\mathcal{G}$, i.e. $\forall \mathcal{G}, \mathcal{G} \preccurlyeq \mathcal{G}$.
- **Functional Equivalence Preservation**: If $\mathcal{G}_1$ is a functional subgraph of $\mathcal{G}_2$, then:
    - (Left-hand Side) if $\mathcal{G}_1'$ is functionally equivalent to $\mathcal{G}_1$, then $\mathcal{G}_1'$ is a functional subgraph of $\mathcal{G}_2$, i.e. if $\mathcal{G}_1 \preccurlyeq \mathcal{G}_2$ and $\mathcal{G}_1' \equiv_{func} \mathcal{G}_1$, then $\mathcal{G}_1' \preccurlyeq \mathcal{G}_2$.
    - (Right-hand Side) if $\mathcal{G}_2'$ is functionally equivalent to $\mathcal{G}_2$, then $\mathcal{G}_1$ is a functional subgraph of $\mathcal{G}_2'$, i.e. if $\mathcal{G}_1 \preccurlyeq \mathcal{G}_2$ and $\mathcal{G}_2' \equiv_{func} \mathcal{G}_2$, then $\mathcal{G}_1 \preccurlyeq \mathcal{G}_2'$.
- **Transitivity**: If $\mathcal{G}_1$ is a functional subgraph of $\mathcal{G}_2$ and $\mathcal{G}_2$ is a functional subgraph of $\mathcal{G}_3$, then $\mathcal{G}_1$ is a functional subgraph of $\mathcal{G}_3$, i.e. if $\mathcal{G}_1 \preccurlyeq \mathcal{G}_2$ and $\mathcal{G}_2 \preccurlyeq \mathcal{G}_3$, then $\mathcal{G}_1 \preccurlyeq \mathcal{G}_3$.
- **Anti-symmetry**: If $\mathcal{G}_1$ is a functional subgraph of $\mathcal{G}_2$, then $\mathcal{G}_2$ is a functional subgraph of $\mathcal{G}_1$ if and only if they are functionally equivalent, i.e. $\mathcal{G}_1 \preccurlyeq \mathcal{G}_2$ and $\mathcal{G}_2 \preccurlyeq \mathcal{G}_1$ if and only if $\mathcal{G}_1 \equiv_{func} \mathcal{G}_2$.

For detailed proofs of the above properties, please refer to Appendix A. It is worth noting that the subgraph equivalence defined in Definition 2 does **not** satisfy the *Transitivity* property. This highlights the improved completeness of the functional subgraph in Definition 3.

## 2.4 Task Definition

Based on Definition 3, we define our primary task:

**Task #1: Functional Subgraph Detection.**   Given a query graph $\mathcal{Q}$ and a candidate graph $\mathcal{G}$, determine if $\mathcal{Q} \preccurlyeq \mathcal{G}$.

While functional subgraph detection is a decision problem (yes/no), it is often desirable to identify **which part** of the target graph $\mathcal{G}$ corresponds to the query function $\mathcal{Q}$. However, as shown in Figure 1, due to potential functional transformations, identifying an exact boundary in the original graph $\mathcal{G}$ that perfectly represents $\mathcal{Q}$ can be challenging or impossible. This leads to our second task, which aims to find the smallest region in $\mathcal{G}$ that encapsulates the function of $\mathcal{Q}$.

**Definition 4** (**Fuzzy Boundary**). *Given a query graph $\mathcal{Q}$ and a candidate graph $\mathcal{G} = (V, E)$, a subgraph $\mathcal{G}^* = (V^*, E^*)$ of $\mathcal{G}$, where $V^* \subseteq V$ and $E^* = E \cap (V^* \times V^*)$, is a **fuzzy boundary** for $\mathcal{Q}$ in $\mathcal{G}$ if:*

1. *$\mathcal{Q} \preccurlyeq \mathcal{G}^*$*

2. *For any proper subgraph $\mathcal{H}$ of $\mathcal{G}^*$ (i.e., $\mathcal{H} \subset \mathcal{G}^*$ and $\mathcal{H} \neq \mathcal{G}^*$), $\mathcal{Q} \not\preccurlyeq \mathcal{H}$*

As illustrated in Figure 1, for $\mathcal{G}$ representing $a \times b + a \times c$ and $\mathcal{Q}$ representing $b + c$, the fuzzy boundary $\mathcal{G}^*$ would likely encompass the components corresponding to $b$, $c$, the two multiplications, and the addition, as this minimal collection is required to functionally contain $b + c$ via transformation. Based on Definition 4, we further define another task as:

**Task #2: Fuzzy Boundary Identification.** Given a query graph $\mathcal{Q}$ and a candidate graph $\mathcal{G}$ such that $\mathcal{Q} \preccurlyeq \mathcal{G}$, determine for each node in $\mathcal{G}$, whether it belongs to the fuzzy boundary $\mathcal{G}^*$ of $\mathcal{Q}$.

## 3 Method

### 3.1 Stage #1: Functional Subgraph Detection

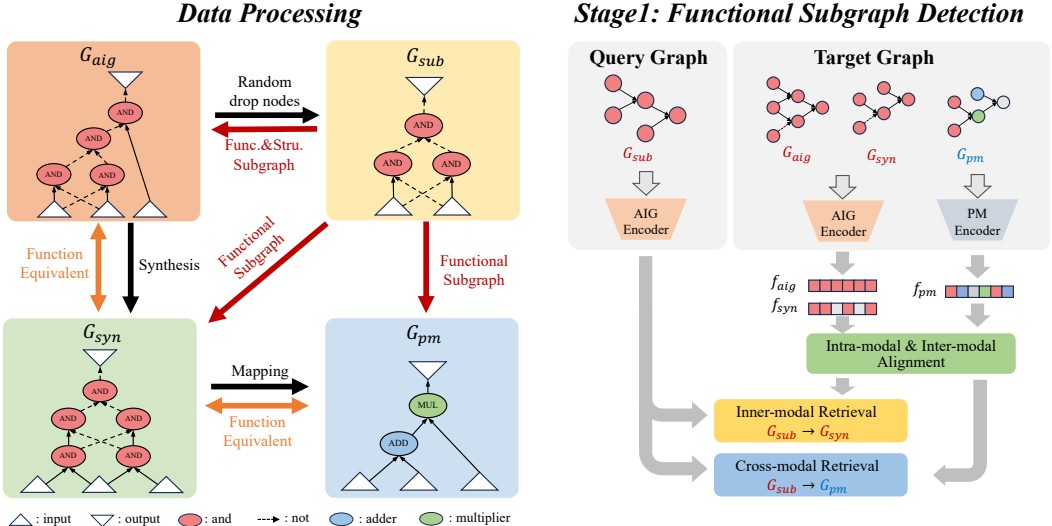

Figure 2: The pipeline of Stage #1. **Left**: Our data processing pipeline. For a given $\mathcal{G}_{aig}$, we first randomly extract a subgraph $\mathcal{G}_{sub}$. Then, we obtain $\mathcal{G}_{syn}$ and $\mathcal{G}_{pm}$ through synthesis and mapping, respectively. **Right**: Our training pipeline via intra-modal and inter-modal alignments for functional subgraph detection. We first encode the query and target graphs using their respective encoders. Next, we perform intra-modal and inter-modal alignment on the target graph to obtain function-invariant and structure-agnostic embeddings. These embeddings are then sent to a task head to determine whether the query graph is contained within the target graph.

**Data Processing** As illustrated in Figure 2, given an AIG netlist $\mathcal{G}_{aig}$, we first randomly drop nodes while ensuring legality, to obtain the subgraph $\mathcal{G}_{sub}$. Next, we use the ABC tool [31] to generate $\mathcal{G}_{syn}$ by randomly selecting a synthesis flow. Importantly, in this step we ensure that $\mathcal{G}_{syn}$ is not isomorphic to $\mathcal{G}_{aig}$. Finally, we apply the ABC tool again to map $\mathcal{G}_{syn}$ to $\mathcal{G}_{pm}$ using the Skywater Open Source PDK [32]. This data processing pipeline ensures that $\mathcal{G}_{aig}$ is equivalent to both $\mathcal{G}_{syn}$ and $\mathcal{G}_{pm}$. Since $\mathcal{G}_{sub}$ is an isomorphic subgraph of $\mathcal{G}_{aig}$, it follows from Definition 3 that $\mathcal{G}_{sub}$ is a functional subgraph of both $\mathcal{G}_{syn}$ and $\mathcal{G}_{pm}$. For negative pairs, following the approach in Li et al. [12], we randomly sample $\mathcal{G}_{aig}$, $\mathcal{G}_{syn}$, and $\mathcal{G}_{pm}$ from other pairs within the same batch. It is important to note that all circuits in this paper have multiple inputs and a single output. For more details, please refer to Section 4.1 and Appendix C.

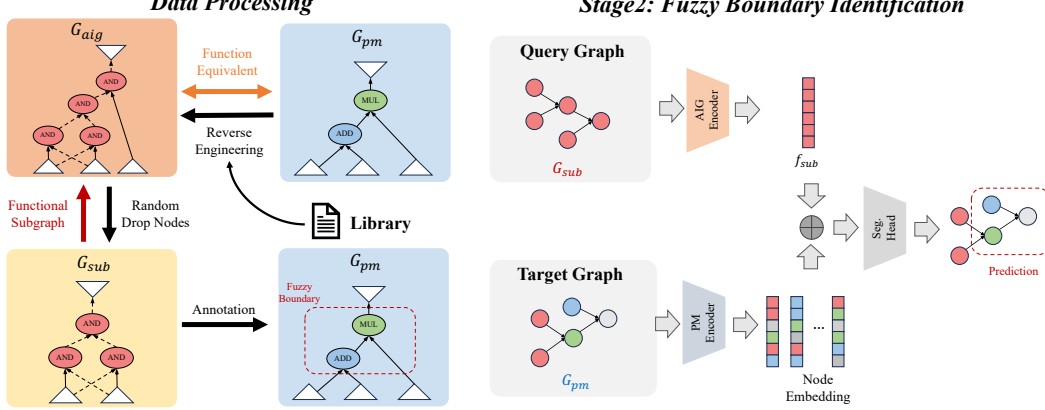

Figure 3: The pipeline of Stage #2. **Left**: Our data processing pipeline. For a given $\mathcal{G}_{pm}$, we replace each node in $\mathcal{G}_{pm}$ with the AIG implementation according to the functionality in the library. Then, we randomly sample a subgraph $\mathcal{G}_{sub}$ from $\mathcal{G}_{aig}$. Finally, we annotate the nodes in $\mathcal{G}_{pm}$ if one of the corresponding AIG nodes still exist in $\mathcal{G}_{sub}$. **Right**: Our training pipeline for fuzzy boundary identification via graph segmentation. Given the query graph $\mathcal{G}_{sub}$ and the target graph $\mathcal{G}_{pm}$, we first use $Enc_{aig}$ to obtain the graph embedding of $\mathcal{G}_{sub}$ and $Enc_{pm}$ to obtain the node embeddings of $\mathcal{G}_{pm}$. These embeddings are then concatenated and passed to a task head to determine whether a node in $\mathcal{G}_{pm}$ lies within the fuzzy boundary of $\mathcal{G}_{sub}$.

**Retrieval** In this paper, we adopt DeepGate2 [22] and DeepCell [25] as backbones for encoding AIG netlists and post-mapping netlists, respectively. Given a query graph $\mathcal{G}_{sub}$, along with positive candidates $\mathcal{G}_{aig}^+, \mathcal{G}_{syn}^+, \mathcal{G}_{pm}^+$ and negative candidates $\mathcal{G}_{aig}^-, \mathcal{G}_{syn}^-, \mathcal{G}_{pm}^-$, we first use the AIG encoder $Enc_{aig}$ and the PM encoder $Enc_{pm}$ for different modalities as follows:

$$f_{sub} = Enc_{aig}(\mathcal{G}_{sub}),\ f_{aig} = Enc_{aig}(\mathcal{G}_{aig})$$
$$f_{syn} = Enc_{aig}(\mathcal{G}_{syn}),\ f_{pm} = Enc_{pm}(\mathcal{G}_{pm})$$

Next, we concatenate the embeddings of the query graph and the candidate graphs and feed them into a classification head, a 3-layer MLP:

$$\hat{y}_{aig} = MLP([f_{sub}, f_{aig}]),\ \hat{y}_{syn} = MLP([f_{sub}, f_{syn}]),\ \hat{y}_{pm} = MLP([f_{sub}, f_{pm}])$$

Finally, we compute the binary cross-entropy (BCE) loss for each prediction:

$$L_{cls} = BCELoss(\hat{y}_{aig}, y_{aig}) + BCELoss(\hat{y}_{syn}, y_{syn}) + BCELoss(\hat{y}_{pm}, y_{pm})$$

**Function-Invariant Alignment** EDA flows such as synthesis and mapping modify the circuit structure while preserving functional equivalence. As defined in Definition 3, the functional subgraph relation focuses on the functionality of the candidate circuits rather than structure, as they can be transformed into an equivalent circuit with any structure. Furthermore, the *Functional Equivalence Preservation* property in Section 2.3 imply that, if the subgraph relation $Q \preccurlyeq G$ hold, then if we replace $Q$ or $G$ with another functional equivalent graph, the subgraph relation continues to hold. This invariance is the key insight for our alignment: the embeddings of functionally equivalent graphs should be aligned, regardless of their structural variations.

Therefore, learning function-invariant embeddings for equivalent circuits across different stages is crucial for functional subgraph detection. While $\mathcal{G}_{aig}$ and $\mathcal{G}_{syn}$ share the same gate types, $\mathcal{G}_{aig}$ and $\mathcal{G}_{pm}$ differ significantly in modality, i.e., the gate types in $\mathcal{G}_{pm}$ are substantially dissimilar to those in $\mathcal{G}_{aig}$. Therefore, we employ both intra-modal and inter-modal alignment techniques to acquire function-invariant and structure-agnostic embeddings with the InfoNCE loss [33]. We select $\mathcal{G}_{aig}$ as the anchor and compute the intra-modal and inter-modal losses as follows:

$$L_{intra} = InfoNCE(f_{aig}^+, f_{syn}^+, f_{syn}^-)$$
$$L_{inter} = InfoNCE(f_{aig}^+, f_{pm}^+, f_{pm}^-)$$

Finally, we summarize the losses for stage #1 as:

$$L_{stage_1} = L_{cls} + L_{intra} + L_{inter}$$

## 3.2 Stage #2: Fuzzy Boundary Identification

**Data Processing**   As illustrated in Figure 3, given a post-mapping netlist $\mathcal{G}_{pm}$, we replace the cells in $\mathcal{G}_{pm}$ with the corresponding AIGs from the library to acquire the netlist $\mathcal{G}_{aig}$. This process yields a mapping function $\phi$ that associates the node indices of $\mathcal{G}_{aig}$ with those of $\mathcal{G}_{pm}$. Next, we randomly drop nodes to obtain $\mathcal{G}_{sub}$, which serves as the functional subgraph of both $\mathcal{G}_{pm}$ and $\mathcal{G}_{aig}$. Using the subgraph $\mathcal{G}_{sub}$, we annotate the nodes in $\mathcal{G}_{pm}$ by mapping the node indices of $\mathcal{G}_{sub}$ to those of $\mathcal{G}_{pm}$ through the function $\phi$. Specifically, for each node in $\mathcal{G}_{sub}$, if it maps to a node $i$ in $\mathcal{G}_{pm}$, we annotate node $i$ as 1; otherwise, we annotate it as 0. This annotation process strictly follows the fuzzy boundary definition in Definition 4.

**Cross-modal Retrieval**   Given a query graph $\mathcal{G}_{sub}$ and a target graph $\mathcal{G}_{pm} = (V_{pm}, E_{pm})$, we first compute the embedding of $\mathcal{G}_{sub}$ and the node embeddings of $\mathcal{G}_{pm}$:

$$f_{sub} = Enc_{aig}(\mathcal{G}_{sub}), \ f_{pm}^1, f_{pm}^2, \ldots, f_{pm}^{|V_{pm}|} = Enc_{pm}(\mathcal{G}_{pm})$$

Next, we use $f_{sub}$ as the query embedding and concatenate it with the node embeddings from $\mathcal{G}_{pm}$. These concatenated embeddings are then fed into a 3-layer MLP for node classification: $\hat{y}_i = MLP([f_{sub}, f_{pm}^i])$. While previous works [13, 29] treat this task as an input-output classification problem, we frame it as a graph segmentation task. This approach arises from the observation that nodes near the input-output nodes contribute to identifying fuzzy boundaries and thus should not be simply labeled as zero. During training, we optimize the model using cross-entropy loss:

$$L_{stage_2} = -\sum_i [y_i \log(\hat{y}_i) + (1 - y_i) \log(1 - \hat{y}_i)] \tag{1}$$

# 4   Experiment

## 4.1   Experimental Setup

We evaluate our method on three AIG datasets: ITC99 [15], OpenABCD [16], and ForgeEDA [17]. Each metric in Tables 1 and 2 is reported as the *mean ± standard deviation* over three independent runs. For data processing, we begin by randomly sampling $k$-hop subgraphs (with $k$ ranging from 8 to 12) to partition large circuits into smaller circuits. Next, we randomly sample subgraphs from these smaller circuits. For logic synthesis, we use the ABC tool [31] with a randomly selected flow from *src_rw*, *src_rs*, *src_rws*, *resyn2rs*, and *compress2rs*. We then apply the VF2 algorithm [6] to verify that the synthesis process has modified the circuit structure. If no modification is detected, we repeat this step until we obtain a circuit with a different structure. For technology mapping, we invoke ABC with the Skywater Open Source PDK [32]. For additional details on the environment, evaluation metrics, and dataset statistics, please refer to Appendix C.

Table 1: Result of Functional Subgraph Detection(%).

| Dataset | Method | $\mathcal{G}_{sub} \to \mathcal{G}_{syn}$ | | | | $\mathcal{G}_{sub} \to \mathcal{G}_{pm}$ | | | |
|---|---|---|---|---|---|---|---|---|---|
| | | Accuracy | Precision | Recall | F1-score | Accuracy | Precision | Recall | F1-score |
| ITC99 | NeuroMatch | $49.8_{\pm0.3}$ | $16.7_{\pm23.6}$ | $33.3_{\pm47.1}$ | $22.2_{\pm31.4}$ | $49.8_{\pm0.2}$ | $16.7_{\pm23.6}$ | $50.0_{\pm50.0}$ | $33.4_{\pm33.4}$ |
| | HGCN | $44.5_{\pm7.7}$ | $35.0_{\pm21.2}$ | $67.3_{\pm46.3}$ | $45.3_{\pm30.2}$ | $49.5_{\pm0.8}$ | $35.7_{\pm20.2}$ | $66.8_{\pm47.0}$ | $44.7_{\pm31.2}$ |
| | Gamora | $50.6_{\pm12.8}$ | $21.1_{\pm27.7}$ | $33.0_{\pm46.0}$ | $25.4_{\pm34.8}$ | $51.7_{\pm4.4}$ | $34.0_{\pm24.2}$ | $51.2_{\pm40.9}$ | $40.2_{\pm29.6}$ |
| | ABGNN | $56.4_{\pm9.1}$ | $20.8_{\pm29.4}$ | $32.7_{\pm46.3}$ | $25.4_{\pm35.9}$ | $54.1_{\pm5.8}$ | $19.0_{\pm26.9}$ | $33.3_{\pm47.1}$ | $24.2_{\pm34.2}$ |
| | Ours | $\mathbf{95.3_{\pm0.1}}$ | $\mathbf{94.4_{\pm0.2}}$ | $\mathbf{96.3_{\pm0.1}}$ | $\mathbf{95.4_{\pm0.0}}$ | $\mathbf{93.1_{\pm0.3}}$ | $\mathbf{92.3_{\pm0.3}}$ | $\mathbf{94.2_{\pm0.9}}$ | $\mathbf{93.2_{\pm0.4}}$ |
| OpenABCD | NeuroMatch | $44.2_{\pm9.8}$ | $17.3_{\pm23.9}$ | $33.4_{\pm47.1}$ | $22.7_{\pm31.8}$ | $44.9_{\pm8.4}$ | $17.0_{\pm23.9}$ | $33.4_{\pm47.1}$ | $22.5_{\pm31.7}$ |
| | HGCN | $52.5_{\pm3.6}$ | $18.0_{\pm25.5}$ | $32.5_{\pm46.0}$ | $23.2_{\pm32.8}$ | $50.0_{\pm0.0}$ | $20.4_{\pm21.4}$ | $33.0_{\pm46.7}$ | $22.2_{\pm31.3}$ |
| | Gamora | $50.8_{\pm1.1}$ | $33.7_{\pm23.9}$ | $66.6_{\pm47.1}$ | $44.8_{\pm31.7}$ | $49.8_{\pm0.3}$ | $33.2_{\pm23.5}$ | $62.1_{\pm44.3}$ | $43.3_{\pm30.6}$ |
| | ABGNN | $34.1_{\pm5.4}$ | $5.2_{\pm3.9}$ | $2.6_{\pm2.6}$ | $3.4_{\pm3.2}$ | $41.3_{\pm4.0}$ | $9.7_{\pm7.6}$ | $3.5_{\pm3.2}$ | $5.1_{\pm4.5}$ |
| | Ours | $\mathbf{92.3_{\pm0.2}}$ | $\mathbf{93.7_{\pm0.2}}$ | $\mathbf{90.6_{\pm0.4}}$ | $\mathbf{92.1_{\pm0.2}}$ | $\mathbf{90.8_{\pm0.4}}$ | $\mathbf{92.4_{\pm0.4}}$ | $\mathbf{88.9_{\pm0.9}}$ | $\mathbf{90.6_{\pm0.5}}$ |
| ForgeEDA | NeuroMatch | $50.0_{\pm0.0}$ | $16.7_{\pm23.6}$ | $33.3_{\pm47.1}$ | $22.2_{\pm31.4}$ | $50.0_{\pm0.0}$ | $16.7_{\pm23.6}$ | $33.3_{\pm47.1}$ | $22.2_{\pm31.4}$ |
| | HGCN | $44.0_{\pm8.5}$ | $18.2_{\pm22.6}$ | $33.9_{\pm46.7}$ | $23.1_{\pm30.8}$ | $48.8_{\pm1.6}$ | $18.8_{\pm22.2}$ | $33.5_{\pm47.0}$ | $22.5_{\pm31.2}$ |
| | Gamora | $40.6_{\pm6.3}$ | $2.4_{\pm1.6}$ | $0.7_{\pm0.8}$ | $1.0_{\pm1.1}$ | $48.2_{\pm1.5}$ | $51.0_{\pm8.2}$ | $31.0_{\pm31.6}$ | $28.5_{\pm22.9}$ |
| | ABGNN | $52.3_{\pm3.3}$ | $34.6_{\pm24.5}$ | $66.6_{\pm47.1}$ | $45.5_{\pm32.2}$ | $52.0_{\pm2.9}$ | $34.4_{\pm24.4}$ | $66.6_{\pm47.1}$ | $45.4_{\pm32.1}$ |
| | Ours | $\mathbf{96.0_{\pm0.1}}$ | $\mathbf{96.8_{\pm0.4}}$ | $\mathbf{95.2_{\pm0.5}}$ | $\mathbf{96.0_{\pm0.1}}$ | $\mathbf{95.3_{\pm0.0}}$ | $\mathbf{95.9_{\pm0.5}}$ | $\mathbf{94.7_{\pm0.5}}$ | $\mathbf{95.3_{\pm0.0}}$ |

## 4.2 Stage #1: Functional Subgraph Detection

We evaluate the performance of our proposed method on three datasets: ITC99, OpenABCD, and ForgeEDA. Our method is compared against several state-of-the-art models, including Neuro-Match [10] and HGCN [12], which are designed for isomorphism subgraph matching in general domain and EDA domain respectively, and Gamora [14] and ABGNN [13], which are designed for reasoning in EDA domain, i.e. for equivalent subgraph matching. Since Gamora and ABGNN focus on boundary detection instead of subgraph matching, we integrate them into the NeuroMatch framework for Stage #1. Further integration of Gamora and ABGNN with our method is discussed in Appendix B. The evaluation metrics include accuracy, precision, recall, and F1-score, computed for two tasks: $\mathcal{G}_{sub}$ to $\mathcal{G}_{syn}$ and $\mathcal{G}_{sub}$ to $\mathcal{G}_{pm}$.

As shown in Table 1, the results on the ITC99, OpenABCD, and ForgeEDA datasets demonstrate that our method significantly outperforms all baseline models. Specifically, for the $\mathcal{G}_{sub} \rightarrow \mathcal{G}_{syn}$ task, our model achieves an average accuracy of $94.5\%$, precision of $95.0\%$, recall of $94.0\%$, and F1-score of $94.5\%$, surpassing all other methods by a large margin. Similarly, for the $\mathcal{G}_{sub} \rightarrow \mathcal{G}_{pm}$ task, our method also shows superior performance with an accuracy of $93.1\%$, precision of $93.5\%$, recall of $92.6\%$, and F1-score of $93.0\%$. In contrast, structure-based methods show an accuracy close to 50% and large standard errors in precision, recall, and F1-score. Such unreliable performance typically arises because these methods indiscriminately predict all pairs as either entirely positive or negative, highlighting their limitations in functional subgraph detection.

## 4.3 Stage #2: Fuzzy Boundary Identification

Table 2: Result of Fuzzy Boundary Identification(%).

| Method | ITC99 | | OpenABCD | | ForgeEDA | |
|---|---|---|---|---|---|---|
| | IoU | DICE | IoU | DICE | IoU | DICE |
| NeuroMatch | $44.2_{\pm 0.0}$ | $61.3_{\pm 0.0}$ | $41.2_{\pm 0.0}$ | $58.3_{\pm 0.0}$ | $42.0_{\pm 0.0}$ | $59.1_{\pm 0.0}$ |
| HGCN | $44.1_{\pm 0.0}$ | $61.2_{\pm 0.0}$ | $41.2_{\pm 0.0}$ | $58.3_{\pm 0.0}$ | $42.0_{\pm 0.0}$ | $59.2_{\pm 0.0}$ |
| Gamora | $39.1_{\pm 2.8}$ | $56.2_{\pm 2.9}$ | $44.2_{\pm 1.2}$ | $61.3_{\pm 1.1}$ | $39.5_{\pm 0.6}$ | $56.6_{\pm 0.6}$ |
| ABGNN | $26.7_{\pm 6.2}$ | $41.7_{\pm 7.5}$ | $37.5_{\pm 0.8}$ | $54.5_{\pm 0.8}$ | $31.9_{\pm 2.6}$ | $48.2_{\pm 3.0}$ |
| Ours | $\mathbf{83.0_{\pm 1.4}}$ | $\mathbf{90.7_{\pm 0.9}}$ | $\mathbf{85.2_{\pm 0.9}}$ | $\mathbf{92.0_{\pm 0.5}}$ | $\mathbf{83.8_{\pm 0.8}}$ | $\mathbf{91.2_{\pm 0.4}}$ |

In this stage, we treat $\mathcal{G}_{sub}$ as the query and aim to locate its fuzzy boundary within the post-mapping netlist $\mathcal{G}_{pm}$. Since Gamora and ABGNN are designed for the detection of the input-output boundary, we first apply each to identify the input and output nodes in $\mathcal{G}_{pm}$. We then perform a BFS between inputs and outputs to recover the corresponding fuzzy boundary, and evaluate the result using Intersection-over-Union (IoU) and DICE score.

Table 2 reports results on ITC99, OpenABCD, and ForgeEDA, demonstrating that our model substantially outperforms all baselines. Specifically, we achieve an average IoU of $\mathbf{84.0}\%$ and a Dice score of $\mathbf{91.3}\%$, significantly outperforming all other methods. Structure-based methods (e.g., NeuroMatch and HGCN) fail to capture functional boundaries and often generate trivial solutions (predicting all nodes as boundary nodes), yielding low variance but poor performance. Although Gamora and ABGNN can detect clear block boundaries for specific arithmetic modules, they struggle with the variable, function-driven fuzzy boundaries required here, resulting in significantly lower performance. Further integration of Gamora and ABGNN within our framework is detailed in Appendix B.

## 4.4 Ablation Study

We perform ablation study on ITC99 dataset and compare the performance of the ablation settings with our proposed method to evaluate the contribution of various components in our method.

**Stage #1 without alignment** achieves accuracy and F1-scores of $94.6\%$ and $94.6\%$ on $\mathcal{G}_{sub} \rightarrow \mathcal{G}_{syn}$ task, which are lower than our method's $95.3\%$ and $95.4\%$. Our model also improves accuracy and F1-score by $1.7\%$ on $\mathcal{G}_{sub} \rightarrow \mathcal{G}_{pm}$ task. These results demonstrate the importance of function-invariant alignment, particularly inter-modal alignment, i.e. aligning $\mathcal{G}_{pm}$ and $\mathcal{G}_{aig}$.

Table 3: Ablation Study on ITC99 Dataset(%).

| Setting | Stage #1 | | | | Stage #2 | |
| --- | --- | --- | --- | --- | --- | --- |
| | $\mathcal{G}_{sub} \to \mathcal{G}_{syn}$ | | $\mathcal{G}_{sub} \to \mathcal{G}_{pm}$ | | $\mathcal{G}_{sub} \to \mathcal{G}_{pm}$ | |
| | Accuracy | F1-score | Accuracy | F1-score | IoU | DICE |
| Stage #1 *wo.* alignment | 94.6 | 94.6 | 91.4 | 91.5 | - | - |
| Stage #2 *wo.* stage #1 | - | - | - | - | 76.3 | 86.5 |
| Stage #2 *wo.* seg. | - | - | - | - | 29.6 | 45.7 |
| Ours | **95.3** | **95.4** | **93.1** | **93.2** | **83.0** | **90.7** |

**Stage #2 without Stage #1** shows a performance drop, with IoU and DICE scores of $76.3\%$ and $86.5\%$, compared to our method's improved values of $83.0\%$ and $90.7\%$. This highlights the crucial role of pretraining knowledge in Stage #1.

**Stage #2 without segmentation** also shows a significant drop in performance, with IoU and DICE values of $29.6\%$ and $45.7\%$, compared to our method's improved $83.0\%$ and $90.7\%$. These results suggest that directly predicting the input-output nodes of the fuzzy boundary is challenging, as it varies with different functional transformations and omits the information of nodes in fuzzy boundary.

## 5   Limitations

While our proposed framework demonstrates strong performance and significant improvements over existing structural approaches, several limitations remain and should be addressed in future research:

**Scalability to Large-scale Circuits:**  Currently, our method has primarily been evaluated on moderately-sized circuits due to computational resource constraints. Real-world EDA applications often involve extremely large netlists with millions of nodes. Scaling our detection and segmentation approaches to handle such large-scale graphs efficiently is non-trivial. Future research could investigate more computationally efficient embedding methods, hierarchical segmentation approaches, or incremental graph processing techniques to enhance scalability.

**Multiple and Overlapping Fuzzy Boundaries:**  Our fuzzy boundary identification method presently assumes a single, minimal enclosing region within the target graph. In practical scenarios, multiple occurrences or overlapping functional subgraphs might exist within a single large circuit, complicating boundary identification tasks. Extending our methodology to effectively handle multiple or overlapping fuzzy boundaries within the same circuit remains an open and challenging direction for further investigation.

**Single-output Circuit Assumption:**  The current approach assumes single-output logic circuits. In real-world scenarios, however, most circuits possess multiple outputs and complex internal functional dependencies. The direct applicability of our method to multi-output circuits, particularly when outputs share significant internal logic, remains unexplored. Generalizing the definitions and embedding strategies to model multi-output scenarios could further enhance practical relevance.

**Non-trivial Function Assumption:**  In this paper, we assume that a graph obtained by removing some nodes and edges is not functionally equivalent to the original graph, i.e. $\forall g \neq \emptyset, G \setminus g \not\equiv_{\text{func}} G$. While EDA tools inherently enforce this constraint, it may limit the generalizability of the functional subgraph in other domains.

By systematically addressing these limitations, subsequent research can extend our approach to broader, more realistic settings, thereby increasing its practical utility in EDA domain and beyond.

## 6   Conclusion

In this paper, we introduce the concept of *functional subgraph matching*, a method to identify implicit logic functions within larger circuits, despite structural variations. We propose a two-stage framework: first, we train models across different modalities with alignment to detect functional subgraphs; second, we fine-tune our model and treat fuzzy boundary identification as a graph segmentation task for precise localization of fuzzy boundary. Evaluations on benchmarks (ITC99, OpenABCD, ForgeEDA) show that our approach outperforms structure-based methods, achieving $93.8\%$ accuracy in functional subgraph detection and a $91.3\%$ DICE score for fuzzy boundary detection.

**Broader Impact** Our method contributes to the advancement of deep learning, particularly in graph-based functional relationship analysis. By improving the detection of functional relationships in complex systems, it has the potential to impact a wide range of applications, from circuit design to other domains that rely on graph functionality, e.g. molecular and protein graphs.

**Acknowledgment** This work was supported in part by the Hong Kong Research Grants Council (RGC) under Grant No. 14212422, 14202824, and C6003-24Y.

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

## A  Proofs of the Proposed Properties

In this section, we use $G_1 \cong G_2$ to denote that $G_1$ is isomorphic to $G_2$. Also, we use $G_1 \equiv_{\text{func}} G_2$ to denote that $G_1$ is functional equivalent to $G_2$.

**Proposition 2.** *If a graph $\mathcal{Q}$ is an equivalent subgraph of $\mathcal{G}$, then $\mathcal{Q}$ is a functional subgraph of $\mathcal{G}$.*

*Proof.* According to the Definition 2, there exists a subgraph $\mathcal{G}'$ of $\mathcal{G}$ such that $\mathcal{Q} \equiv_{func} \mathcal{G}'$. By replacing $\mathcal{G}'$ with $\mathcal{Q}$, we get $\bar{\mathcal{G}} = \mathcal{G} \setminus \mathcal{G}' \cup \mathcal{Q}$ which is equivalent to $\mathcal{G}$ and a subgraph of $\mathcal{G}$ is isomorphic to $\mathcal{Q}$. By the Definition 3, $\mathcal{Q}$ is a functional subgraph of $\mathcal{G}$. $\square$

**Proposition 3** (Reflexivity). $\forall \mathcal{G}, \mathcal{G} \preccurlyeq \mathcal{G}$.

*Proof.* $\mathcal{G}$ is a subgraph of itself, and $\mathcal{G} \equiv_{\text{func}} \mathcal{G}$. By the definition of functional subgraph, it follows that $\mathcal{G} \preccurlyeq \mathcal{G}$. $\square$

**Proposition 4** (Functional Equivalence Preservation). *If $\mathcal{G}_1$ is a functional subgraph of $\mathcal{G}_2$, then:*

- *(Left-hand Side) if $\mathcal{G}_1 \preccurlyeq \mathcal{G}_2$ and $\mathcal{G}'_1 \equiv_{func} \mathcal{G}_1$, then $\mathcal{G}'_1 \preccurlyeq \mathcal{G}_2$.*
- *(Right-hand Side) if $\mathcal{G}_1 \preccurlyeq \mathcal{G}_2$ and $\mathcal{G}'_2 \equiv_{func} \mathcal{G}_2$, then $\mathcal{G}_1 \preccurlyeq \mathcal{G}'_2$.*

*Proof.* **(Right-hand Side)** According to the Definition 3, if $\mathcal{G}_1$ is a functional subgraph of $\mathcal{G}_2$, then there exist $\mathcal{G}^*_2$ that $\mathcal{G}^*_2 \equiv_{\text{func}} \mathcal{G}_2$ and $\mathcal{G}_1$ is an isomorphic subgraph of $\mathcal{G}^*_2$. Since $\mathcal{G}'_2 \equiv_{\text{func}} \mathcal{G}_2$ and $\mathcal{G}^*_2 \equiv_{\text{func}} \mathcal{G}_2$, then $\mathcal{G}^*_2$. Since $\mathcal{G}'_2 \equiv_{\text{func}} \mathcal{G}'_2$. By the Definition 3, $\mathcal{G}_1$ is a functional subgraph of $\mathcal{G}'_2$.

**(Left-hand Side)** By Definition 3, there exists a graph $\mathcal{G}'_2 \equiv_{\text{func}} \mathcal{G}_2$, such that

$$\mathcal{G}_1 \cong \bar{\mathcal{G}}_2, \tag{2}$$

where $\bar{\mathcal{G}}_2$ is a subgraph of $\mathcal{G}'_2$. Since $\mathcal{G}_1 \cong \bar{\mathcal{G}}_2$, it follows that

$$\mathcal{G}_1 \equiv_{\text{func}} \bar{\mathcal{G}}_2. \tag{3}$$

By the transitivity of functional equivalence, we then have

$$\mathcal{G}_1 \equiv_{\text{func}} \bar{\mathcal{G}}_2 \equiv_{\text{func}} \mathcal{G}'_1. \tag{4}$$

Thus, by replacing $\bar{\mathcal{G}}_2$ in $\mathcal{G}'_2$ with $\mathcal{G}'_1$, we obtain a new graph

$$\mathcal{G}''_2 = (\mathcal{G}'_2 \setminus \bar{\mathcal{G}}_2) \cup \mathcal{G}'_1, \tag{5}$$

which satisfies

$$\mathcal{G}''_2 \equiv_{\text{func}} \mathcal{G}_2. \tag{6}$$

From the definition of functional equivalence, we know that $\mathcal{G}''_2 \equiv_{\text{func}} \mathcal{G}_2$ and that $\mathcal{G}'_1$ is a subgraph of $\mathcal{G}''_2$. Therefore, it follows that

$$\mathcal{G}'_1 \preccurlyeq \mathcal{G}_2. \tag{7}$$

$\square$

**Proposition 5** (Transitivity). *If $\mathcal{G}_1 \preccurlyeq \mathcal{G}_2$ and $\mathcal{G}_2 \preccurlyeq \mathcal{G}_3$, then $\mathcal{G}_1 \preccurlyeq \mathcal{G}_3$.*

*Proof.* By definition, there exists a graph $\mathcal{G}'_2 \equiv_{\text{func}} \mathcal{G}_2$, such that

$$\mathcal{G}_1 \cong \bar{\mathcal{G}}_1, \text{ and } \bar{\mathcal{G}}_1 \text{ is a subgraph of } \mathcal{G}'_2. \tag{8}$$

Since $\mathcal{G}'_2 \equiv_{\text{func}} \mathcal{G}_2$, by Proposition 4, it follows that $\mathcal{G}'_2 \preccurlyeq \mathcal{G}_3$.

Therefore, there exists a graph $\mathcal{G}'_3 \equiv_{\text{func}} \mathcal{G}_3$, and $\mathcal{G}'_2$ is a subgraph of $\mathcal{G}'_3$. Since $\bar{\mathcal{G}}_1$ is a subgraph of $\mathcal{G}'_2$ and $\mathcal{G}'_2$ is a subgraph of $\mathcal{G}'_3$, it follows that $\bar{\mathcal{G}}_1$ is a subgraph of $\mathcal{G}'_3$.

Since $\mathcal{G}_1 \cong \bar{\mathcal{G}}_1$, $\mathcal{G}'_3 \equiv_{\text{func}} \mathcal{G}_3$ and $\bar{\mathcal{G}}_1$ is a subgraph of $\mathcal{G}'_3$, by the definition of functional subgraph, we conclude that

$$\mathcal{G}_1 \preccurlyeq \mathcal{G}_3. \tag{9}$$

$\square$

**Proposition 6** (Anti-symmetry). *$\mathcal{G}_1 \preccurlyeq \mathcal{G}_2$ and $\mathcal{G}_2 \preccurlyeq \mathcal{G}_1$ if and only if $\mathcal{G}_1 \equiv_{\text{func}} \mathcal{G}_2$.*

*Proof.* ($\Rightarrow$) Since $\mathcal{G}_1 \preccurlyeq \mathcal{G}_2$, we have

$$\mathcal{G}_1 \cong \mathcal{G}'_2 \setminus g, \quad \text{and} \quad \mathcal{G}_2 \equiv_{\text{func}} \mathcal{G}'_2. \tag{10}$$

Since $\mathcal{G}_2 \preccurlyeq \mathcal{G}_1$, we have $\mathcal{G}'_2 \preccurlyeq \mathcal{G}'_2 \setminus g$. By the definition of functional subgraphs, there exists a graph $\mathcal{G}_3$ such that $\mathcal{G}_3 \equiv_{\text{func}} \mathcal{G}'_2 \setminus g$ and $\mathcal{G}'_2$ is a subgraph of $\mathcal{G}_3$. This implies that $\mathcal{G}'_2 \cong \mathcal{G}_3 \setminus g'$, so we also have

$$\mathcal{G}'_2 \equiv_{\text{func}} \mathcal{G}_3 \setminus g'. \tag{11}$$

Since $\mathcal{G}_3 \equiv_{\text{func}} \mathcal{G}'_2 \setminus g$, it follows that

$$\mathcal{G}_3 \cup g \equiv_{\text{func}} \mathcal{G}'_2. \tag{12}$$

Thus, we have

$$\mathcal{G}_3 \cup g \equiv_{\text{func}} \mathcal{G}'_2 \equiv_{\text{func}} \mathcal{G}_3 \setminus g'. \tag{13}$$

Note that in Section 2.3, we assume that a graph obtained by removing some nodes and edges is not functionally equivalent to the original graph, i.e., $\forall g \neq \emptyset$, $G \setminus g \not\equiv_{\text{func}} G$. Therefore, we must have $g = g' = \emptyset$, which implies

$$\mathcal{G}_1 \cong \mathcal{G}'_2 \setminus g \cong \mathcal{G}'_2, \quad \text{and} \quad \mathcal{G}_2 \equiv_{\text{func}} \mathcal{G}'_2. \tag{14}$$

Thus, we conclude that

$$\mathcal{G}_1 \equiv_{\text{func}} \mathcal{G}_2. \tag{15}$$

($\Leftarrow$) If $\mathcal{G}_1 \equiv_{\text{func}} \mathcal{G}_2$, since $\mathcal{G}_1 \preccurlyeq \mathcal{G}_1$ and $\mathcal{G}_2 \preccurlyeq \mathcal{G}_2$, according to *Functional Equivalence Preservation* property, it follows that $\mathcal{G}_1 \preccurlyeq \mathcal{G}_2$ and $\mathcal{G}_2 \preccurlyeq \mathcal{G}_1$. $\square$

# B   Additional Experimental Results

## B.1   Functional Subgraph Matching

Considering that the encoder in our method can be replaced with other backbones, we test our approach with different encoders and propose baselines for the functional subgraph detection task, as shown in Table 4.

## B.2   Fuzzy Boundary Identification

We further evaluate these methods on fuzzy boundary identification. The results are shown in Table 5.

Table 5: Result of baselines in stage #2.

| Method | ITC99 | | OpenABCD | | ForgeEDA | |
|---|---|---|---|---|---|---|
| | IoU | DICE | IoU | DICE | IoU | DICE |
| Ours+Gamora | 82.1 | 90.2 | 81.4 | 89.8 | 83.6 | 91.1 |
| Ours+ABGNN | 82.7 | 90.5 | 84.4 | 91.5 | **88.4** | **93.8** |
| Ours | **83.0** | **90.7** | **85.2** | **92.0** | 83.8 | 91.2 |

Table 4: Result of baselines in stage #1.

| Dataset | Method | $\mathcal{G}_{sub} \to \mathcal{G}_{syn}$ | | | | $\mathcal{G}_{sub} \to \mathcal{G}_{pm}$ | | | |
|---|---|---|---|---|---|---|---|---|---|
| | | Accuracy | Precision | Recall | F1-score | Accuracy | Precision | Recall | F1-score |
| ITC99 | Ours+Gamora | 90.8 | 91.1 | 90.4 | 90.7 | 86.4 | 88.6 | 83.5 | 86.0 |
| | Ours+ABGNN | 87.9 | 83.1 | 95.1 | 88.7 | 88.2 | 82.8 | **96.5** | 89.1 |
| | Ours | **95.3** | **94.4** | **96.3** | **95.4** | **93.1** | **92.3** | 94.2 | **93.2** |
| OpenABCD | Ours+Gamora | 90.1 | 89.6 | **90.7** | 90.2 | **91.0** | 89.3 | **93.2** | **91.2** |
| | Ours+ABGNN | 81.7 | 78.5 | 87.5 | 82.7 | 83.3 | 78.9 | 91.1 | 84.5 |
| | Ours | **92.3** | **93.7** | 90.6 | **92.1** | 90.8 | **92.4** | 88.9 | 90.6 |
| ForgeEDA | Ours+Gamora | 94.2 | 95.9 | 92.4 | 94.1 | 80.6 | 93.8 | 65.5 | 77.1 |
| | Ours+ABGNN | 89.7 | 88.5 | 91.2 | 89.8 | 87.6 | 88.3 | 86.8 | 87.5 |
| | Ours | **96.0** | **96.8** | **95.2** | **96.0** | **95.3** | **95.9** | **94.7** | **95.3** |

## B.3 Scalability on Medium-Sized Circuits

we collect a medium-sized graph dataset from ForgeEDA [17], containing circuits with graph sizes ranging from 100 to 10000 nodes. The statistical information of the medium-sized dataset is shown in Table 6.

Table 6: Statistics of the medium-sized dataset.

| Graph Type | Nodes | Edges | Depth |
|---|---|---|---|
| $G_{sub}$ | $192.1 \pm 320.87$ | $207.16 \pm 354.01$ | $27.91 \pm 30.08$ |
| $G_{syn}$ | $1396.99 \pm 1764.63$ | $1958.84 \pm 2519.23$ | $66.48 \pm 99.7$ |
| $G_{pm}$ | $679.93 \pm 851.2$ | $1352.96 \pm 1703.16$ | $21.83 \pm 32.6$ |

We perform functional subgraph detection on this dataset, and the results are shown in Table 7 and 8. Since ABGNN encounters out-of-memory error when encoding graphs with deep logic levels, we do not report its results on this new dataset. While our method still demonstrates state-of-the-art performance, it shows a significant performance drop (from an F1-score of 95.3% to 81.2%, as shown in Table 1). This result highlights the challenge of scaling to larger circuits. We hope future work will explore and address this challenge.

Table 7: Functional Subgraph Detection on $G_{sub} \to G_{syn}$.

| Method | Accuracy | Precision | Recall | F1-score |
|---|---|---|---|---|
| NeuroMatch | $51.2_{\pm 3.3}$ | $34.6_{\pm 24.5}$ | $66.7_{\pm 47.1}$ | $45.5_{\pm 32.2}$ |
| HGCN | $50.0_{\pm 0.0}$ | $16.7_{\pm 23.6}$ | $33.3_{\pm 47.1}$ | $22.2_{\pm 31.4}$ |
| Gamora | $50.0_{\pm 0.0}$ | $40.0_{\pm 14.1}$ | $66.7_{\pm 47.1}$ | $44.6_{\pm 31.3}$ |
| **Ours** | $\mathbf{81.5_{\pm 0.6}}$ | $\mathbf{82.7_{\pm 1.1}}$ | $\mathbf{79.8_{\pm 1.7}}$ | $\mathbf{81.2_{\pm 0.8}}$ |

## B.4 Comparison with VF3

we evaluate the state-of-the-art subgraph isomorphic heuristic algorithm VF3 [34], which consistently achieves 100% precision and 100% recall on standard subgraph isomorphism tasks. Due to time constraints, we sampled circuits with fewer than 50 nodes from the ForgeEDA dataset and applied the VF3 algorithm. The results are shown in Table 9. According to our Definition 3 of Functional Subgraph, if $Q$ is an isomorphic subgraph of $G$, then $Q$ is always a functional subgraph of $G$. This is demonstrated by the 100% precision achieved by VF3. However, due to the function-preserving transformation, the explicit structure of $Q$ often disappears, leading to extremely low recall(0.32%) for VF3. These results highlight the importance of task definition.

Table 8: Functional Subgraph Detection on $G_{sub} \rightarrow G_{pm}$.

| Method | Accuracy | Precision | Recall | F1-score |
|---|---|---|---|---|
| NeuroMatch | $51.0_{\pm 1.2}$ | $33.9_{\pm 24.0}$ | $66.6_{\pm 47.1}$ | $44.9_{\pm 31.8}$ |
| HGCN | $50.0_{\pm 0.0}$ | $16.7_{\pm 23.6}$ | $33.3_{\pm 47.1}$ | $22.2_{\pm 31.4}$ |
| Gamora | $50.0_{\pm 0.0}$ | $33.3_{\pm 23.6}$ | $66.7_{\pm 47.1}$ | $44.4_{\pm 31.4}$ |
| **Ours** | $\mathbf{78.9_{\pm 1.0}}$ | $\mathbf{80.6_{\pm 1.6}}$ | $\mathbf{76.3_{\pm 2.7}}$ | $\mathbf{78.3_{\pm 1.3}}$ |

Table 9: Comparison of subgraph isomorphism methods on different tasks.

| Method | Runtime (s) | $G_{sub} \rightarrow G_{syn}$ | | | $G_{sub} \rightarrow G_{pm}$ | | |
|---|---|---|---|---|---|---|---|
| | | Precision | Recall | F1-score | Precision | Recall | F1-score |
| VF3 | 480.8 | **100.0** | 0.32 | 0.65 | — | — | — |
| **Ours** | **8.0** | 88.5 | **91.9** | **90.2** | **86.4** | **94.5** | **90.3** |

## C Datasets and Implementation Details

**Dataset** Dataset statistics and splits are shown in Table 10. For dataset split, we first split the training circuits and test circuits in the source dataset, then we cut subgraph for the training circuit and test circuits to generate our small circuit dataset. For ITC99 and OpenABCD, the split follow the previous work [24]. For ForgeEDA, we randomly select $10\%$ circuits in the dataset as test circuits. For small circuit, we apply Algorithm 1 to randomly sample subgraph.

---

**Algorithm 1** Random Sample Subgraph

---

**Input:** ndoes $\mathcal{V}$, edges $\mathcal{E}$, root $r$
**Output:** nodes $V$, edges $E$, root $r$
    Build adjacency $G$ from $\mathcal{E}$
    **if** $rand(0,1) < 0.5$ **then**
        Set $r$ to the predecessor $p \in G[r]$ that maximizes $\text{predCount}(p)$
    **end if**
    $\rho \leftarrow rand(0.6, 0.95)$
    $T \leftarrow \rho \cdot |\mathcal{V}|, V \leftarrow \{r\}, Q \leftarrow [r], E \leftarrow \emptyset$
    **while** $Q \neq \emptyset \wedge |V| < T$ **do**
      $n \leftarrow \text{pop}(Q)$
      **for all** $v \in \text{shuffle}(G[n])$ **do**
        **if** $v \notin V$ **then**
          $\text{push}(Q, v); V \cup \{v\}; E \cup \{(n, v)\}$
        **end if**
      **end for**
    **end while**
    **return** $V, E, r$

---

Table 10: Dataset Statistics. We report average and standard error with $avg. \pm std$.

| Source Dataset | Split | #Pair | $\mathcal{G}_{sub}$ | | $\mathcal{G}_{aig}$ | | $\mathcal{G}_{syn}$ | | $\mathcal{G}_{pm}$ | |
|---|---|---|---|---|---|---|---|---|---|---|
| | | | #Node | Depth | #Node | Depth | #Node | Depth | #Node | Depth |
| ITC99 | train | 36592 | $248_{\pm 132}$ | $15.0_{\pm 2.0}$ | $320_{\pm 166}$ | $19.1_{\pm 3.0}$ | $315_{\pm 164}$ | $19.0_{\pm 3.0}$ | $179_{\pm 91}$ | $6.9_{\pm 1.0}$ |
| | test | 5917 | $218_{\pm 113}$ | $14.0_{\pm 2.0}$ | $282_{\pm 141}$ | $17.3_{\pm 2.2}$ | $278_{\pm 138}$ | $17.0_{\pm 2.0}$ | $157_{\pm 79}$ | $6.3_{\pm 0.9}$ |
| OpenABCD | train | 54939 | $155_{\pm 113}$ | $13.0_{\pm 2.0}$ | $203_{\pm 140}$ | $16.4_{\pm 3.2}$ | $198_{\pm 134}$ | $16.0_{\pm 3.0}$ | $108_{\pm 75}$ | $5.8_{\pm 1.1}$ |
| | test | 9726 | $100_{\pm 66}$ | $13.0_{\pm 2.0}$ | $132_{\pm 84}$ | $16.0_{\pm 2.2}$ | $128_{\pm 82}$ | $15.0_{\pm 2.0}$ | $69_{\pm 46}$ | $5.5_{\pm 0.9}$ |
| ForgeEDA | train | 60183 | $126_{\pm 102}$ | $13.4_{\pm 3.5}$ | $161_{\pm 129}$ | $16.6_{\pm 4.2}$ | $156_{\pm 125}$ | $16.2_{\pm 4.5}$ | $88_{\pm 69}$ | $5.8_{\pm 1.4}$ |
| | test | 7753 | $127_{\pm 96}$ | $13.6_{\pm 3.3}$ | $163_{\pm 122}$ | $17.0_{\pm 3.8}$ | $159_{\pm 120}$ | $16.4_{\pm 4.2}$ | $89_{\pm 65}$ | $5.9_{\pm 1.3}$ |

**Environment**    All experiments are run on an NVIDIA A100 GPU with 64 GB of memory. Models are trained using the Adam optimizer with a learning rate of 0.001, a batch size of 1024. We train our model in stage#1 for 100 epochs and finetune it in stage#2 for 10 epochs. Training our model on one dataset takes approximately 10 hours. Model architectures follow the configurations specified in the original works except that we set the hidden dimension to 128 for all models.

**Evaluation Metrics**    For Stage #1, we measure classification performance by accuracy and report precision, recall and f1-score according to the counts of true positives (TP), true negatives (TN), false positives (FP), and false negatives (FN):

$$\text{Precision} = \frac{\text{TP}}{\text{TP} + \text{FP}}, \ \text{Recall} = \frac{\text{TP}}{\text{TP} + \text{FN}},$$
$$\text{Accuracy} = \frac{\text{TP} + \text{TN}}{\text{TP} + \text{TN} + \text{FP} + \text{FN}}, \ \text{F1-score} = \frac{2 \times \text{Precision} \times \text{Recall}}{\text{Precision} + \text{Recall}}.$$

For Stage #2, which is similar to a segmentation task, we use Intersection over Union (IoU) and the Dice coefficient. Let $P$ be the set of predicted positive nodes and $G$ the set of ground-truth positive nodes:

$$\text{IoU} = \frac{|P \cap G|}{|P \cup G|}, \ \text{Dice} = \frac{2\,|P \cap G|}{|P| + |G|} \tag{16}$$

# D    Background

**And-Inverter Graph(AIG)**    In our works, AIG is a directed acyclic graph (DAG) composed of three basic elements: Primary Input(PI), AND gate and NOT gate. For example, a simple logic expression $\neg A \wedge B$ can be build as a DAG with 2 PIs(A and B), one NOT gate and one AND gate. The edges are $[(A, NOT), (NOT, AND), (B, AND)]$. Since the out-degree of $AND$ is zero, it represents the final output.

**Technology Mapping**    Technology Mapping is a function-preserving transformation that converts an AIG into a post-mapping (PM) netlist. While the AIG consists of simple logic elements, such as AND and NOT gates, the basic components in a PM netlist can be more complex, such as adders or multipliers. As a result, node types can differ significantly between the two forms, and this is why we consider AIG and PM netlists as distinct modalities in this paper.

**Logic Synthesis**    Logic synthesis aims to simplify the structure of an AIG while preserving its functionality. It transforms one AIG into another with a simpler structure. For example, the expression $eq_1 : (A \wedge B) \wedge (A \wedge C)$ can be simplified to $eq_2 : A \wedge B \wedge C$. Although $eq_1$ and $eq_2$ are functional equivalent, $eq_2$ uses only 2 AND gates compared to 3 AND gates in $eq_1$.

**InfoNCE Loss**    InfoNCE (Information Noise-Contrastive Estimation) is a contrastive loss used in self-supervised learning. Its goal is to identify a single "positive" sample from a set of "negative" samples for a given "anchor" sample. It pulls the anchor and positive representations together while pushing the anchor and negatives apart:

$$\mathcal{L}_{InfoNCE} = -\log \frac{\exp(\text{sim}(q, k_+)/\tau)}{\sum_{i=0}^{N} \exp(\text{sim}(q, k_i)/\tau)} \tag{17}$$

where $q$ is the anchor, $k_+$ is the positive, $k_i$ are the negatives, sim is a similarity function (like dot product), and $\tau$ is a temperature hyperparameter.

