# OpenReview forum: "Functional Matching of Logic Subgraphs: Beyond Structural Isomorphism"
_NeurIPS.cc/2025/Conference — NeurIPS 2025 poster_

### Official Review · Reviewer_tTFV · 2025-06-18

**Clarity:** 3
**Significance:** 2
**Originality:** 2
**Rating:** 4
**Confidence:** 3

**Summary:**

The paper presents a novel concept of functional subgraph matching, enabling the identification of implicit logic functions within larger circuits. The proposed two-stage process, which combines intra-modal and inter-modal alignment with graph segmentation for boundary identification, offers a unique perspective on addressing circuit optimization, verification, and security analysis challenges in EDA applications, and has achieved best subgraph matching accuracy in multiple tasks.

**Questions:**

1. Please list the scale of each evaluated dataset, show the runtime and correpsonding computational cost of each method
2. Comparing to more different subgraph matching algorithms instead of GNNs
3. Add more experiments on medium to large graph datasets.

**Ethical Concerns:**

["NO or VERY MINOR ethics concerns only"]

**Final Justification:**

The author's rebuttal is insightful, and the paper is above the acceptance threshold.

**Limitations:**

See questions

**Quality:**

2

**Strengths And Weaknesses:**

The paper presents a novel concept of functional subgraph matching, enabling the identification of implicit logic functions within larger circuits. The proposed two-stage process, which combines intra-modal and inter-modal alignment with graph segmentation for boundary identification, offers a unique perspective on addressing circuit optimization, verification, and security analysis challenges in EDA applications, and has achieved best subgraph matching accuracy in multiple tasks.
Weaknesses:
1. Experiment scale is not enough
2. subgraph matching application is limited
3. Experiments comparisons limited to GNNs, but less on other subgraph matching methods.

---

> ### Author Rebuttal · Authors · 2025-07-31
>
> We thank the reviewer for valuable feedback. Here are our answers to the weaknesses and questions.
>
> **Q1:Please list the scale of each evaluated dataset, show the runtime and corresponding computational cost of each method**
>
> A1: Thank you for the suggestion. The scale of each evaluated dataset is provided in Table 6 in Appendix. In addition, we report the runtime and GPU memory consumption for each method across all datasets:
>
> | |NeuroMatch|HGCN|Gamora|ABGNN|Ours|
> |-|-:|-:|-:|-:|-:|
> |ITC99 runtime(s)|3.7|2.1|2.3|3.1|28.3|
> |ITC99 memory usage(GB)|4.0|0.7|0.4|0.4|1.1|
> |ITC99 F1-Score(%)|22.2|45.3|25.4|25.4|95.4|
> |OpenABCD runtime(s)|3.9|2.4|2.4|3.3|29.3|
> |OpenABCD memory usage(GB)|3.3|0.6|0.4|0.4|0.9|
> |OpenABCD F1-Score(%)|22.7|23.2|44.8|3.4|92.1|
> |ForgeEDA runtime(s)|4.4|2.5|2.3|3.2|36.4|
> |ForgeEDA memory usage(GB)|6.9|1.2|0.8|0.7|1.8|
> |ForgeEDA F1-Score(%)|22.2|23.1|1.0|45.5|96.0|
>
>
>
> The runtime and memory usage primarily rely on the GNN architecture. While our method costs a longer runtime compared to other learning-based approaches, we want to emphasize that **optimizing GNN architecture for the runtime or memory usage is not the focus of this work**. Our primary contribution lies in introducing the novel concept of functional subgraph matching and proposing a method that can effectively identify the functional subgraph, while previous works struggle with unreliable performance.
>
> **Q2:Comparing to more different subgraph matching algorithms instead of GNNs**
>
> A2: To the best of our knowledge, we are the first to propose the concept of the functional subgraph and formally define the tasks of functional subgraph detection and fuzzy boundary identification. As a result, **no existing subgraph matching algorithms are directly applicable to functional subgraph matching**.
>
> We provide a Python-style pseudocode for a naive algorithm based on our definition:
>
> ```
> def functional_subgraph_matching(Q,G):
>     """
>     Q:Query graph
>     G: Candidate graph
>     """
>     candidate_list = find_all_functional_equivalent_graph(G)
>     for G_prime in candidate_list:
>         if is_subgraph_isomorphic(Q,G_prime):
>             return True
>     return False
> ```
> In this algorithm, `is_subgraph_isomorphic(Q,G)` can be implemented with heuristic algorithm like VF3[1]. However, the key challenge lie in  `find_all_functional_equivalent_graph(G)`. For example, in the case of an n-bit prefix adder, the number of possible designs is  $\frac{1}{n}\tbinom{2(n-1)}{n-1}$, which implies that the complexity for find all equivalent graphs is $O(\frac{4^n}{n^{3/2}})$. Therefore, it's impossible to list all the functional equivalent graph for a circuit, and thus we cannot compare with this naive algorithm. This highlights the necessity of designing heuristic algorithms for this process.
>
> Moreover, directly applying heuristics on structural subgraph isomorphism to functional subgraph matching leads to many failure cases, as discussed in Section 2. We further conducted an empirical study to illustrate this limitation.
>
> To evaluate heuristics, we sample circuits with less than 50 nodes in ForgeEDA dataset, and only apply the state-of-the-art heuristic algorithms, VF3[1], since other algorithm could be extremely time-costing as discussed in previous works[2]. The result is list below:
> | Method | Runtime (s) | Precision (sub→syn) | Recall (sub→syn) | F1-score (sub→syn) | Precision (sub→pm) | Recall (sub→pm) | F1-score (sub→pm) |
> |-|-:|-:|-:|-:|-:|-:|-:|
> | VF3  | 480.8 | 100 | 0.32 | 0.65 | –   | –   | –    |
> | Ours | 8.0  | 88.5 | 91.9 | 90.2 | 86.4 | 94.5 | 90.3 |
>
> According to our Definition 3 of Functional Subgraph, if $Q$ is a isomorphic subgraph of $G$, then $Q$ is always a functional subgraph of $G$. This is demonstrated by the 100% precision achieved by VF3. However, due to the function-preserving transformation, the explicit structure of $Q$ often disappear, leading to extremely low recall(0.32%) for VF3.
>
> *[1] Carletti et al., “VF3: Challenging the Time Complexity of Exact Subgraph Isomorphism,” *IEEE TPAMI*, 2018.*
>
> *[2] Lou et al., “Neural Subgraph Matching,” *arXiv*, 2020.*
>
> **Q3:Experiment scale is not enough. Add more experiments on medium to large graph datasets**
>
>
> A3: Thank you for your suggestion. When scaling to large circuits, we categorize the tasks into two cases:
> 1. The target is a large circuit, and the query is also a large or medium circuit
> 2. The target is a large circuit, while the query is a small circuit
>
> For the first case, the bottleneck arises from the functional diversity of large circuits. A small dataset cannot sufficiently cover the wide range of possible designs, leading to out-of-distribution scenarios during testing. For example, a circuit with $n$ input and $1$ output could present up to $2^{2^n}$ different functions, making it extremely difficult to capture all functional variations in very large circuits (e.g., those with millions of gates).
>
> To explore this setting, we collect a medium-sized graph dataset from ForgeEDA, containing circuits with graph sizes ranging from 100 to 10000 nodes. The statistic information of the medium-sized dataset is shown below.
>
> | |Nodes|Edges|Depth|
> |-|-|-|-|
> |$G_{sub}$|$192.1\pm320.87$|$207.16\pm354.01$|$27.91\pm30.08$|
> |$G_{syn}$|$1396.99\pm1764.63$|$1958.84\pm2519.23$|$66.48\pm99.7$|
> |$G_{pm}$|$679.93\pm851.2$|$1352.96\pm1703.16$|$21.83\pm32.6$|
>
>
> We perform functional subgraph detection on this dataset, and the results are shown in Table 1 and 2. Since ABGNN encounters out-of-memory error when encoding graphs with deep logic levels, we do not report its results on this new dataset.
>
> Table 1. Functional Subgraph Detection on $G_{sub} \to G_{syn}$
>
> |Method|Accuracy|Precision|Recall|F1-score|
> |-|-|-|-|-|
> |NeuroMatch|$51.2_{\pm3.3}$|$34.6_{\pm24.5}$|$66.7_{\pm47.1}$|$45.5_{\pm32.2}$|
> |HGCN|$50.0_{\pm0.0}$|$16.7_{\pm23.6}$|$33.3_{\pm47.1}$|$22.2_{\pm31.4}$|
> |Gamora|$50.0_{\pm0.0}$|$40.0_{\pm14.1}$|$66.7_{\pm47.1}$|$44.6_{\pm31.3}$|
> |Ours|$81.5_{\pm0.6}$|$82.7_{\pm1.1}$|$79.8_{\pm1.7}$|$81.2_{\pm0.8}$|
>
>
>
> Table 2. Functional Subgraph Detection on $G_{sub} \to G_{pm}$
>
> |Method|Accuracy|Precision|Recall|F1-score|
> |-|-|-|-|-|
> |NeuroMatch|$51.0_{\pm1.2}$|$33.9_{\pm24.0}$|$66.6_{\pm47.1}$|$44.9_{\pm31.8}$|
> |HGCN|$50.0_{\pm0.0}$|$16.7_{\pm23.6}$|$33.3_{\pm47.1}$|$22.2_{\pm31.4}$|
> |Gamora|$50.0_{\pm0.0}$|$33.3_{\pm23.6}$|$66.7_{\pm47.1}$|$44.4_{\pm31.4}$|
> |Ours|$78.9_{\pm1.0}$|$80.6_{\pm1.6}$|$76.3_{\pm2.7}$|$78.3_{\pm1.3}$|
>
>
> While our method still demonstrates SOTA performance, it shows a significant performance drop. This result highlights the challenge of scaling to larger circuits. We hope future work will explore and address this challenge.
>
> For the second case, the bottleneck stems from the extreme size imbalance between the query and the candidate circuit. A candidate may contain millions of gates, while the query contains only a few hundred, creating a highly imbalanced learning problem. Despite prior works[3] in the object detection domain have proposed various strategies to address this type of imbalance, it remains a significant challenge in the graph domain.
>
> However, in practice, when the target circuit is very large, a **divide-and-conquer strategy** is commonly adopted to transform the small–large retrieval task into multiple small–small retrieval tasks. For example, in the case of large sequential circuits, we can partition the design through flip-flops to get much smaller combinational subcircuits. Similarly, if the query circuit is a k-hop graph, HGCN[4] cuts the candidate circuit into  2k-hop subgraphs for subgraph matching. This approach aligns with the original setting of our work.
>
> We further summarize the scale of datasets used in prior works and our study in the table below. In fact, due to the np-complete nature of subgraph isomorphism, most of methods on subgraph isomorphism works on small datasets. For example, COX2, DD, MSRC_21, FIRSTMMDB, ENZYMES and SYNTHETIC are datasets used in NeuroMatch. 2-hop,3-hop,4-hop and 5-hop are datasets used in HGCN. And the remaining datasets are used in our works.
>
> |Dataset|Target Size (nodes)|Target Depth|Query Size (nodes)|Query Depth|
> |--|--:|--:|--:|--:|
> |COX2|41.6|-|22.4|-|
> |DD|30.0|-|17.8|-|
> |MSRC_21|79.6|-|22.6|-|
> |FIRSTMMDB|30.0|-|17.8|-|
> |ENZYMES|35.7|-|17.4|-|
> |SYNTHETIC|30.2|-|17.5|-|
> |2-hop|-|4|15.65|2|
> |3-hop|-|6|40.5|3|
> |4-hop|-|8|73.1|4|
> |5-hop|-|10|180.5|5|
> |ITC99|315.4|17|248.0|15|
> |OpenABCD|198.1|16|155.0|13|
> |ForgeEDA|156.5|16.4|126.0|13.6|
>
>
> *[1] Fey et al., “GNN-AutoScale: Scalable and Expressive Graph Neural Networks via Historical Embeddings,” *ICML*, 2021.*
>
> *[2] Zheng et al., “DeepGate4: Efficient and Effective Representation Learning for Circuit Design at Scale,” *ICLR*, 2025.*
>
> *[3] Cheng et al., “Towards Large-Scale Small Object Detection: Survey and Benchmarks,” *IEEE TPAMI*, 2023.*
>
> *[4] Li et al., “Efficient Subgraph Matching for Fast Subcircuit Identification,” *MLCAD*, 2024.*
>
>
> **Q4:subgraph matching application is limited**
>
> A4: We acknowledge the reviewer's concern and would like to emphasize that functional subgraph matching **has broader application on circuit optimization, verification, and security analyses**, as discussed on our Introduction Section.
>
> In addition to these applications, our proposed functional subgraph can also be applied to other tasks. For example, in the Engineering Change Order (ECO)[1,2], a critical process of hardware development, there are situations circuit designers would need to extract complicated arithmetic circuitry deeply embedded inside a fully optimized flattened netlist without the knowing of module boundary and IO positions. The extracted results can then be used for replacement by more advanced intellectual property (IP) macros.
>
> *[1] Wei et al., “Universal Macro Block Mapping for Arithmetic Circuits,” *DATE*, 2015.*
>
> *[2] Diao et al., “Coupling Reverse Engineering and SAT for Arithmetic Circuit Verification,” *ASP-DAC*, 2016.*

---

> > ### Comment · Reviewer_tTFV · 2025-08-01
> > **Acknowledgement**
> >
> > The contribution and novelty are good now, please add additional experiments into final version, and add the explanation of find_all_functional_equivalent_graph more clearly.

---

> > > ### Author Response · Authors · 2025-08-03
> > >
> > > We appreciate the reviewer’s recognition of our contribution and the novelty of our work. We will include the additional experiment in the final version.
> > >
> > > Regarding the `find_all_functional_equivalent_graph`, given a circuit graph, there are nearly infinite functionally equivalent circuits. Without any constraints, it is difficult to perform a specific analysis. Therefore, we can only provide examples under explicit constraints (e.g., an n-bit prefix adder).
> > >
> > > We further present two cases based on basic function-preserving transformations. We analyze the design space under each transformation separately, without considering other types of transformations:
> > >
> > > - **Commutative Property**: The commutative property of circuits states that $A \land B = B \land A$. Given a circuit of the form $x_1 \land x_2 \land \dots \land x_n$, the number of possible permutations of the variables is $O(n!)$.
> > > - **Associative Property**: The associative property states that $A \land B \land C = (A \land B) \land C = A \land (B \land C) = (A \land C) \land B$. Applying this property to circuits like $x_1 \land x_2 \land \dots \land x_n$ is a classical problem: finding the design space of binary trees with $n$ leaves. The size of this design space is given by the Catalan number $C_{n-1} = \frac{1}{n}\binom{2(n-1)}{n-1}$.
> > >
> > > These two simple examples demonstrate **how quickly the design space of functionally equivalent circuits can grow**. While we only discuss two basic properties, incorporating more advanced properties, such as $(A \land B) \land (A \land C) = A \land B \land C$, would further expand the design space, turning it into a complex combinatorial problem. This complexity makes it hard to give a comprehensive analysis of the full space.

---

### Official Review · Reviewer_WZEu · 2025-06-30

**Clarity:** 2
**Significance:** 4
**Originality:** 3
**Rating:** 5
**Confidence:** 3

**Summary:**

The work is situated in the context of electronic design automation. The task they focus on is subgraph matching: detecting whether/where a query graph is present within a larger target graph. They 1) propose an improved definition of this task that also covers transformations on the target graph, essentially more broadly identifying functionally equivalent subgraphs; and 2) propose a two-stage deep learning framework to solve this problem. The empirical evaluation indicates significant improvement compared to prior work.

**Questions:**

What part of the framework is causing the strong results compared to the competitors (or why do the competitors perform poorly)? Do I understand correctly that they have not been designed to consider functional subgraphs; only subgraph isomorphism or subgraph equivalence, and that partially explains it? Would the competitors perform as well as your method if there were no transformations and the setting was instead a subgraph isomorphism/subgraph equivalence task?

**Ethical Concerns:**

["NO or VERY MINOR ethics concerns only"]

**Final Justification:**

I confirm having read the other reviews and thank the authors for the rebuttal.

I remain positive about the work for the previously outlined points, and have slightly increased my confidence.

The main new idea/insight of the paper (the proposed new task of identifying functional subgraphs) is well motivated and poses a useful contribution to the field. The empirical results are strong (although there are challenges in scaling to much larger graphs) and the work is presented clearly.

**Limitations:**

yes

**Quality:**

3

**Strengths And Weaknesses:**

**Clarity:**

The work is well structured and includes clear informative figures that help guide the reading process. On a high level, the contributions are very clear. On a more detailed level, I am not sure I understood the deep learning pipeline. For example, what does "intra-modal and inter-modal alignment" mean exactly? Are these just the losses of line 201?. Related, more generally, the work often discusses how, but not the why; which may contribute to a reader not understanding why something works.


The work does assume some familiarity with the field: concepts such as netlist, "And-Inverter Graph", and "technology mapping" are not introduced. However, even without familiarity it is easy to understand the contributions on a high level.



Further comments
* Figure 3 is never referred to.

* Would be good to define the InfoNCE loss function (in the appendix)



**Originality:** To the best of my knowledge, the work is very original.


**Significance:**

To work is well motivated. In particular, the proposed new task (identifying functional subgraphs) makes a lot of sense, and the empirical results are very strong.
The work, which uses a neural pipeline, is relevant to the conference.


**Quality:**
The work appears sound and the empirical setting appears thorough, demonstrating strong results. As a limitation, they do only test on smaller circuits, and do not yet consider scaling to large-scale graphs. This is correctly discussed in the limitations section.

---

> ### Author Rebuttal · Authors · 2025-07-31
>
> We thank the reviewer for valuable feedback. Here are our answers to the weaknesses and questions.
>
> **Q1:What part of the framework is causing the strong results compared to the competitors? Would the competitors perform as well as your method if there were no transformations and the setting was instead a subgraph isomorphism/subgraph equivalence task?**
>
> A1: Regarding the performance increasing, we provide a more detailed discussion below. Overall, we attribute the improvement in performance to three parts:
> * Task definition
> * Two-stage framework
> * Function learning ability of GNN model
>
>
> **Task Definition**
> To the best of our knowledge, our work introduces the concept of a functional subgraph for the first time, which differs significantly from subgraph isomorphism and subgraph equivalence, as discussed in Section 2. This difference in definition naturally leads to a different task formulation.
>
> To further illustrate this, we evaluate the state-of-the-art subgraph isomorphic heuristic algorithm VF3 [1], which consistently achieves 100% precision and 100% recall on standard subgraph isomorphism tasks. Due to time constraints, we sampled circuits with fewer than 50 nodes from the ForgeEDA dataset and applied the VF3 algorithm. The results are shown below:
>
> |Method|Runtime (s)|Precision (sub→syn)|Recall (sub→syn)|F1-score (sub→syn)|Precision (sub→pm)|Recall (sub→pm)|F1-score (sub→pm)|
> |-|-:|-:|-:|-:|-:|-:|-:|
> |VF3|480.8|100|0.32|0.65|–|–|–|
> |Ours|8|88.5|91.9|90.2|86.4|94.5|90.3|
>
> According to our Definition 3 of Functional Subgraph, if $Q$ is an isomorphic subgraph of $G$, then $Q$ is always a functional subgraph of $G$. This is demonstrated by the 100% precision achieved by VF3. However, due to the function-preserving transformation, the explicit structure of $Q$ often disappear, leading to extremely low recall(0.32%) for VF3. This result highlight the importance of task definition.
>
>
>
>
> **Two-Stage Framework**
> Table 4 in Section 4.4 discusses the effectiveness of our proposed alignment, two-stage training and graph segmentation. The results demonstrate the effectiveness of each component in our method.
>
> Specifically, for fuzzy boundary identification, previous works on targeting arithmetic block identification (Subgraph Equivalence), such as Gamora [2] and ABGNN [3], focus on identifying exact input/output patterns. When identifying a half adder, Gamora, for instance, requires the root of the subgraph to be a XOR gate. However, after function-preserving transformations, such explicit boundaries may disappear (as discussed in Section 2.4), leading to significant performance degradation for these methods.
>
> Furthermore, considering that the graph encoder in our method can be replaced with other backbones, we evaluate our approach using alternative encoders designed for function-related tasks and propose several baselines, as shown in Tables 4 and 5 in Appendix B. The results demonstrate that with our proposed framework, these models gain significant performance improvement, indicating the importance of our proposed two-stage framework.
>
>
> **Function learning ability of GNN model**
> Prior works on subgraph isomorphism mainly focus on designing GNN models to learn structural patterns rather than functional behavior. As a result, these models perform poorly on function-related tasks, as discussed in prior studies [4,5].
>
> For function-related method, like Gamora and ABGNN, we have integrate these graph encoder into our framework, and the results are as shown in Tables 4 and 5 in Appendix B. Even though with a significant performance improvement, these method still shows a significant performance drop compared to our method, indicating that the GNN architecture also leading to performance gap.
>
> *[1]Carletti et al., “VF3: Challenging the Time Complexity of Exact Subgraph Isomorphism,” IEEE TPAMI, 2018.*
>
> *[2] Wu et al., “Gamora: Graph Learning-Based Symbolic Reasoning for Large-Scale Boolean Networks,” DAC, 2023.*
>
> *[3] Wang et al., “Efficient Arithmetic Block Identification with Graph Learning and Network-Flow,” IEEE TCAD, 2022.*
>
> *[4] Liu et al., “Polargate: Breaking the Functionality Representation Bottleneck of AND-Inverter Graph Neural Network,” ICCAD, 2024.*
>
> *[5] Zheng et al., “DeepGate4: Efficient and Effective Representation Learning for Circuit Design at Scale,” ICLR, 2025.*
>
> **Q2: The work does assume some familiarity with the field: concepts such as netlist, "And-Inverter Graph", and "technology mapping" are not introduced.**
>
> A2: Thank you for pointing this out. We will add a section in the Appendix to introduce the background concepts for better clarity. Below is a brief introduction to the key terms used in our work:
> * **And-Inverter Graph(AIG)**: In our works, AIG is a directed acyclic graph (DAG) composed of three basic elements: Primary Input(PI), AND gate and NOT gate. For example, a simple logic expression $\lnot A\land B$ can be build as a DAG with 2 PIs(A and B), one NOT gate and one AND gate. The edges are $[(A,NOT), (NOT,AND),(B,AND)]$. Since the out-degree of $AND$ is zero, it represents the final output.
> * **Technology Mapping**: This is a function-preserving transformation that converts an AIG into a post-mapping (PM) netlist. While the AIG consists of simple logic elements, such as AND and NOT gates, the basic components in a PM netlist can be more complex, such as adders or multipliers. As a result, node types can differ significantly between the two forms, and this is why we consider AIG and PM netlists as distinct modalities in this paper.
> * **Logic Synthesis**: Logic synthesis aims to simplify the structure of an AIG while preserving its functionality. It transforms one AIG into another with a simpler structure. For example, the expression $eq_1: (A\land B)\land(A\land C)$ can be simplified to $eq_2:A\land B \land C$.  Although $eq_1$ and $eq_2$ are functional equivalent, $eq_2$ uses only 2 AND gates compared to 3 AND gates in $eq_1$.
>
>
> **Q3: What does "intra-modal and inter-modal alignment" mean exactly? ... the work often discusses how, but not the why**
>
> A3: As discussed in Q2, we treat AIG and PM netlists as different modalities due to their differing node types. In our work, we consider two types of function-preserving transformations: Logic Synthesis and Technology Mapping.
>
> Given an AIG $G_{aig}$, we apply Logic Synthesis to obtain $G_{syn}$, and Technology Mapping to obtain the PM netlist $G_{pm}$. All three graphs are functionally equivalent.
>
> We align the embeddings of these functionally equivalent graphs to ensure that their representations remain the same across transformations. This alignment is performed in two ways:
>
> - **Intra-modal alignment**: Aligning embeddings between graphs within the same modality, i.e. between $G_{aig}$ and $G_{syn}$, both of which are AIGs.
> - **Inter-modal alignment**: Aligning embeddings across different modalities, i.e. between $G_{aig}$ (an AIG) and $G_{pm}$ (a PM netlist).
>
> Before discussing the details of alignment, we would like to clarify its motivation.
>
>
> Let's first go back to the definition of Functional Subgraph:
>
> **Definition(Functional Subgraph)**: A graph $Q$ is a functional subgraph of $G$, denoted $Q \preccurlyeq G$, if there exists a graph $G'$ such that $G' \equiv_{func} G$ and $Q$ is isomorphic to a subgraph of $G'$.
>
> According to the definition, if $Q\preccurlyeq G_1$ and $G_1\equiv_{func}G_2$, then $Q\preccurlyeq G_2$. This denotes that the subgraph relation is preserved under functional equivalence **on the right-hand side** of the relation $\preccurlyeq$.
>
> In addition, we proposed a property, Functional Equivalence Preservation, in Section 2.3.
>
> **Proposition(Functional Equivalence Preservation)**: If $G_1$ is a functional subgraph of $G_2$, and $G'_1$ is functionally equivalent to $G_1$, then $G'_1$ is a functional subgraph of $G_2$.
>
> This property shows that the subgraph relation is preserved under functional equivalence **on the left-hand side** of the relation $\preccurlyeq$.
>
> These two results imply that, if the subgraph relation $Q\preccurlyeq G$ hold, then if we replace $Q$ or $G$ with another functional equivalent graph, the subgraph relation continues to hold. This invariance is the key motivation for our alignment: the embeddings of functionally equivalent graphs should be aligned, regardless of their structural variations.
>
> The equation in line 201 in our paper shows the details of alignment. Since the modality of PM-netlist is different from AIG, we divide the alignment into two parts: (1)intra-modal alignment: AIG-AIG embedding alignment (2)inter-modal  alignment: AIG-PM embedding alignment.
>
> Specifically, given embedding $[f_{aig}^+,f_{syn}^+,f_{syn}^-]$, $f_{aig}^+$ and $f_{syn}^+$ are functional equivalent, thus, following the idea of contrastive learning,  we aim to minimize the distance of $f_{aig}^+$ and $f_{syn}^+$. Similarly, $f_{aig}^+$ and $f_{syn}^-$ are functional unequivalent, thus we aim to maximize the distance of $f_{aig}^+$ and $f_{syn}^-$. For $[f_{aig}^+,f_{pm}^+,f_{pm}^-]$, we apply the same contrastive objective.
>
> This approach allows us to enforce functional similarity in the learned embedding space, which is essential for robust functional subgraph detection across structural transformations.
>
> **Q4: Figure 3 is never referred to. Would be good to define the InfoNCE loss function (in the appendix)**
>
> A4: Thanks for your suggestion. We will revise the final version to refer to Figure 3 in the main text and include a formal definition of the InfoNCE loss function in the appendix for completeness.

---

> > ### Comment · Reviewer_WZEu · 2025-08-04
> >
> > Thank you for the answers; I remain positive about the work and have slightly increased my confidence.

---

### Official Review · Reviewer_Ytof · 2025-06-30

**Clarity:** 3
**Significance:** 2
**Originality:** 3
**Rating:** 4
**Confidence:** 4

**Summary:**

This paper introduces a novel approach called functional subgraph matching to identify logic subgraphs in circuits based on functionality rather than structural isomorphism, addressing challenges posed by synthesis transformations that alter circuit topology. The proposed two-stage multi-modal framework combines functional embeddings and graph segmentation techniques to detect functional subgraphs and identify fuzzy boundaries with high accuracy. Experimental results on standard benchmarks demonstrate significant improvements over traditional structure-based methods

**Questions:**

Q1. What are the main computational bottlenecks currently preventing scalability to industry-size netlists, and what concrete modifications would be necessary to handle graphs with millions of nodes?

**Ethical Concerns:**

["NO or VERY MINOR ethics concerns only"]

**Final Justification:**

The rebuttal address my concerns on the assumption and theoretical analysis.

**Limitations:**

Yes

**Quality:**

3

**Strengths And Weaknesses:**

S1.  The problem studied in the paper is interesting.
S2.  A two stage algorithm is proposed to address this problem.
S3. Experiments are conducted to evaluate the proposed algorithms.


W1. As stated in Section 5, results are limited to moderately sized circuits. The methodology's ability to scale to very large circuits (millions of gates) — common in industry — is not empirically demonstrated, and concrete plans for achieving this remain high-level.
W2. The assumption that any node/edge removal changes the circuit's function may not always hold, especially outside the benchmark datasets or when circuit simplification is possible but not legal in ABC.
W3. While the definitions are carefully formalized and properties stated, there is a lack of theoretical analysis on complexity, guarantees, or failure modes of the learning-based approach, especially compared to existing exact/approximate subgraph methods in graph ML.

---

> ### Author Rebuttal · Authors · 2025-07-31
>
> We thank the reviewer for valuable feedback. Here are our answers to the weaknesses and questions.
>
> **Q1: What are the main computational bottlenecks currently preventing scalability to industry-size netlists?**
>
> A1: In fact, computational complexity is not the primary bottleneck. Prior works have proposed training strategies that enable GNN models to scale to graphs with millions of nodes [1,2]. The real bottleneck lies in the nature of the task itself. When scaling to large circuits, we categorize the tasks into two cases:
> 1. The target is a large circuit, and the query is also a large or medium circuit
> 2. The target is a large circuit, while the query is a small circuit
>
> For the first case, the bottleneck arises from the functional diversity of large circuits. A small dataset cannot sufficiently cover the wide range of possible designs, leading to out-of-distribution (OOD) scenarios during testing. For example, a circuit with $n$ input and $1$ output could present up to $2^{2^n}$ different functions, making it extremely difficult to capture all functional variations in very large circuits (e.g., those with millions of gates).
>
> To explore this setting, we collect a medium-sized graph dataset from ForgeEDA, containing circuits with graph sizes ranging from 100 to 10000 nodes. The statistical information of the medium-sized dataset is shown below.
>
> | |Nodes|Edges|Depth|
> |-|-|-|-|
> |$G_{sub}$|$192.1\pm320.87$|$207.16\pm354.01$|$27.91\pm30.08$|
> |$G_{syn}$|$1396.99\pm1764.63$|$1958.84\pm2519.23$|$66.48\pm99.7$|
> |$G_{pm}$|$679.93\pm851.2$|$1352.96\pm1703.16$|$21.83\pm32.6$|
>
>
>
> We perform functional subgraph detection on this dataset, and the results are shown in Table 1 and 2. Since ABGNN encounters out-of-memory error when encoding graphs with deep logic levels, we do not report its results on this new dataset.
>
> Table 1. Functional Subgraph Detection on $G_{sub} \to G_{syn}$
>
> | Method | Accuracy | Precision | Recall | F1-score |
> |-|-|-|-|-|
> | NeuroMatch | $51.2_{\pm3.3}$ | $34.6_{\pm24.5}$ | $66.7_{\pm47.1}$ | $45.5_{\pm32.2}$ |
> | HGCN | $50.0_{\pm0.0}$ | $16.7_{\pm23.6}$ | $33.3_{\pm47.1}$ | $22.2_{\pm31.4}$ |
> | Gamora | $50.0_{\pm0.0}$ | $40.0_{\pm14.1}$ | $66.7_{\pm47.1}$ | $44.6_{\pm31.3}$ |
> | Ours | $81.5_{\pm0.6}$ | $82.7_{\pm1.1}$ | $79.8_{\pm1.7}$ | $81.2_{\pm0.8}$ |
>
>
>
> Table 2. Functional Subgraph Detection on $G_{sub} \to G_{pm}$
>
> | Method | Accuracy | Precision | Recall | F1-score |
> |-|-|-|-|-|
> | NeuroMatch | $51.0_{\pm1.2}$ | $33.9_{\pm24.0}$ | $66.6_{\pm47.1}$ | $44.9_{\pm31.8}$ |
> | HGCN | $50.0_{\pm0.0}$ | $16.7_{\pm23.6}$ | $33.3_{\pm47.1}$ | $22.2_{\pm31.4}$ |
> | Gamora | $50.0_{\pm0.0}$ | $33.3_{\pm23.6}$ | $66.7_{\pm47.1}$ | $44.4_{\pm31.4}$ |
> | Ours | $78.9_{\pm1.0}$ | $80.6_{\pm1.6}$ | $76.3_{\pm2.7}$ | $78.3_{\pm1.3}$ |
>
> While our method still demonstrates state-of-the-art performance, it shows a significant performance drop (from an F1-score of 95.3% to 81.2%, as shown in Table 1). This result highlights the challenge of scaling to larger circuits. We hope future work will explore and address this challenge.
>
> For the second case, the bottleneck stems from the extreme size imbalance between the query and the candidate circuit. A candidate may contain millions of gates, while the query contains only a few hundred. In such scenarios, the query may occupy as little as 0.001% of the candidate, creating a highly imbalanced learning problem. Despite prior works[3] in the object detection domain have proposed various strategies to address this type of imbalance, it remains a significant challenge in the graph domain.
>
> However, in practice, when the target circuit is very large, a divide-and-conquer strategy is commonly adopted to transform the small–large retrieval task into multiple small–small retrieval tasks. For example, in the case of large sequential circuits, we can partition the design through flip-flops to get much smaller combinational subcircuits. Similarly, HGCN [4] also divide large candidate circuits into smaller subgraphs for independent retrieval. Specifically, if the query circuit is a k-hop graph, HGCN cuts the candidate circuit into  2k-hop subgraphs for subgraph matching. This approach aligns with the original setting of our work.
>
>
> *[1]Fey M, Lenssen J E, Weichert F, et al. Gnnautoscale: Scalable and expressive graph neural networks via historical embeddings[C]//International conference on machine learning. PMLR, 2021: 3294-3304.*
>
> *[2]Zheng Z, Huang S, Zhong J, et al. DeepGate4: Efficient and Effective Representation Learning for Circuit Design at Scale[C]//The Thirteenth International Conference on Learning Representations.*
>
> *[3] Cheng G, Yuan X, Yao X, et al. Towards large-scale small object detection: Survey and benchmarks[J]. IEEE transactions on pattern analysis and machine intelligence, 2023, 45(11): 13467-13488.*
>
> *[4]Li B, Wang S, Chen T, et al. Efficient subgraph matching
> for fast subcircuit identification[C]//Proceedings of the 2024 ACM/IEEE International Symposium on Machine Learning for CAD. 2024: 1-7.*
>
> **Q2: The assumption that any node/edge removal changes the circuit's function may not always hold, especially outside the benchmark datasets or when circuit simplification is possible but not legal in ABC.**
>
> A2: We acknowledge that Non-trivial Function Assumption may not always hold in other cases, as discussed in Section 5, and it may limit the generalizability of our proposed functional subgraph and corresponding model to other domains.
>
> The assumption of a non-trivial function is made because, for example, if we consider a function $f(x)$, adding one NOT gate results in $\lnot f(x)$, and adding two NOT gate results in $\lnot \lnot f(x)$. According to our definition, this leads to the result: $f(x)\preccurlyeq \lnot f(x)\preccurlyeq \lnot \lnot f(x) \equiv_{func} f(x)$, which could cause confusion on definition.
>
> Therefore, theoretically, if there exist trivial function in a circuit, we suggest removing these function since it won't impact the circuit's functionality and the subgraph relation, as discussed in Functional Equivalence Preservation property in Section 2.3.
>
> Practically, a better approach would be to send the circuit to ABC to ensure the assumption holds, and then transform it into a graph structure.
>
> We also hope the future works could address this limitation by introducing a concept with improved completeness.
>
> **Q3: While the definitions are carefully formalized and properties stated, there is a lack of theoretical analysis on complexity, guarantees, or failure modes of the learning-based approach, especially compared to existing exact/approximate subgraph methods in graph ML.**
>
> A3: We appreciate the reviewer’s recognition of our proposed definition and properties. As our method relies on deep learning, providing formal theoretical analysis on guarantees or failure modes is inherently challenging, which is also a common limitation in the deep learning field.
>
> However, we conducted empirical evaluations of runtime and memory usage across different methods. The results are summarized below:
>
>
>
> | | NeuroMatch | HGCN | Gamora | ABGNN | Ours |
> |-|-:|-:|-:|-:|-:|
> | ITC99 runtime(s) | 3.7 | 2.1 | 2.3 | 3.1 | 28.3 |
> | ITC99 memory usage(GB) | 4.0 | 0.7 | 0.4 | 0.4 | 1.1 |
> | ITC99 F1-Score(%) | 22.2 | 45.3 | 25.4 | 25.4 | 95.4 |
> | OpenABCD runtime(s) | 3.9 | 2.4 | 2.4 | 3.3 | 29.3 |
> | OpenABCD memory usage(GB) | 3.3 | 0.6 | 0.4 | 0.4 | 0.9 |
> | OpenABCD F1-Score(%) | 22.7 | 23.2 | 44.8 | 3.4 | 92.1 |
> | ForgeEDA runtime(s) | 4.4 | 2.5 | 2.3 | 3.2 | 36.4 |
> | ForgeEDA memory usage(GB) | 6.9 | 1.2 | 0.8 | 0.7 | 1.8 |
> | ForgeEDA F1-Score(%) | 22.2 | 23.1 | 1.0 | 45.5 | 96.0 |
>
>
> We further evaluated the runtime on heuristics. For the heuristics, such as VF2[1] and RI[2] algorithm, are costly and can sometimes take unexpectedly long time(hours), even for relatively small queries (of size 20), as discussed in previous works[3]. Therefore, to evaluate heuristics, we sample circuits with less than 50 nodes in Forgeeda dataset, and only apply state-of-the-art heuristic, VF3[4] algorithm. VF3 required 480.8 seconds, while our method took 8 seconds, demonstrating that heuristics can be significantly slower than learning-based approaches.
>
> |Method|Runtime (s)|Precision (sub→syn)|Recall (sub→syn)|F1-score (sub→syn)|Precision (sub→pm)|Recall (sub→pm)|F1-score (sub→pm)|
> |-|-|-|-|-|-|-|-|
> |VF3|480.8|100|0.32|0.65|–|–|–|
> |Ours|8|88.5|91.9|90.2|86.4|94.5|90.3|
>
>
>
>
> While our method costs a longer runtime compared to other learning-based approaches, we want to emphasize that **optimizing GNN architecture for the runtime or memory usage is not the focus of this work**. Our primary contribution lies in introducing the novel concept of functional subgraph matching and proposing a method that can effectively identify the functional subgraph, while previous works(even heuristics like VF3) struggle with unreliable performance.
>
> *[1]Cordella L P, Foggia P, Sansone C, et al. A (sub) graph isomorphism algorithm for matching large graphs[J]. IEEE transactions on pattern analysis and machine intelligence, 2004, 26(10): 1367-1372.*
>
> *[2]Bonnici V, Giugno R, Pulvirenti A, et al. A subgraph isomorphism algorithm and its application to biochemical data[J]. BMC bioinformatics, 2013, 14(Suppl 7): S13.*
>
> *[3] Lou Z, You J, Wen C, et al. Neural subgraph matching[J]. arXiv preprint arXiv:2007.03092, 2020.*
>
> *[4]Challenging the time complexity of exact subgraph isomorphism for huge and dense graphs with VF3 - Carletti V., Foggia P., Saggese A., Vento M. - IEEE transactions on pattern analysis and machine intelligence - 2018*

---

> ### Comment · Area_Chair_29Qu · 2025-08-05
>
> Dear Reviewer,
>
> This is a kind reminder that the discussion phase will be ending soon on Aug 8th. Please read the author's responses and engage in a constructive discussion with the authors.
>
> Thank you for your time and cooperation.
>
> Best,
>
> NeurIPS 2025 Area Chair

---

### Official Review · Reviewer_apyh · 2025-07-03

**Clarity:** 3
**Significance:** 2
**Originality:** 3
**Rating:** 4
**Confidence:** 3

**Summary:**

This paper proposes a functional subgraph matching method for subgraph matching  in EDA, which shifts focus from brittle structural matching to logic-aware functionality—critical for real-world EDA. The proposed method include 2 stages: the first stage trains multimodal embeddings (intra/inter-modal alignment) for cross-stage functional detection, while the second stage formulates boundary identification as graph segmentation (not classification) to improve continuity/accuracy. Experiments EDA benchmarks demonstrates the proposed method significantly outperforms baselines.

**Questions:**

More discussion about the performance increase would be preferred.

**Ethical Concerns:**

["NO or VERY MINOR ethics concerns only"]

**Limitations:**

Yes.

**Paper Formatting Concerns:**

N.A.

**Quality:**

3

**Strengths And Weaknesses:**

Strengths:
1. This paper is well motivated and is easy to follow.
2. The experiment results is solid and proposed method significantly boosts the DICE score on EDA benchmarks (ITC99, OpenABCD, ForgeEDA), far exceeding structural baselines.
3. Directly enables cross-stage verification (e.g., from pre-synthesis to post-layout).

Weaknesses
Overall, I think the novelty of the proposed method is limited in some sense. From my perspective, the key idea of this paper is a combination of subgraph classification, DVAE[1], and CktGNN[2]. Overall, the key concept is to find the functional equivalence rather than structural isomorphism. More discussion about this concern is preferred.

[1] Zhang M, Jiang S, Cui Z, et al. D-vae: A variational autoencoder for directed acyclic graphs[J]. Advances in neural information processing systems, 2019, 32.
[2] Dong Z, Cao W, Zhang M, et al. CktGNN: Circuit Graph Neural Network for Electronic Design Automation[C]//The Eleventh International Conference on Learning Representations.

---

> ### Author Rebuttal · Authors · 2025-07-31
>
> We thank the reviewer for valuable feedback. Here are our answers to the weaknesses and questions.
>
> **Q1: More discussion about the performance increase would be preferred.**
>
> A1: Regarding the performance increase, we provide more detailed discussion below. Overall, we attribute our performance improvement to 3 parts:
> * Task definition
> * Two-stage framework
> * Function learning ability of GNN model
>
>
> **Task Definition**
> To the best of our knowledge, our work introduces the concept of a functional subgraph for the first time, which differs significantly from subgraph isomorphism and subgraph equivalence, as discussed in Section 2. This difference in definition naturally leads to a different task formulation.
>
> To further illustrate this, we evaluate the state-of-the-art subgraph isomorphic heuristic algorithm VF3 [1], which consistently achieves 100% precision and 100% recall on standard subgraph isomorphism tasks. Due to time constraints, we sampled circuits with fewer than 50 nodes from the ForgeEDA dataset and applied the VF3 algorithm. The results are shown below:
>
> | Method | Runtime (s) | Precision (sub→syn) | Recall (sub→syn) | F1-score (sub→syn) | Precision (sub→pm) | Recall (sub→pm) | F1-score (sub→pm) |
> |--------|-----------:|--------------------:|-----------------:|-------------------:|-------------------:|----------------:|------------------:|
> | VF3    | 480.8     | 100.0              | 0.32             | 0.65               | –                  | –               | –                 |
> | Ours   | 8.0        | 88.5               | 91.9             | 90.2               | 86.4               | 94.5            | 90.3              |
>
> According to our Definition 3 of Functional Subgraph, if $Q$ is an isomorphic subgraph of $G$, then $Q$ is always a functional subgraph of $G$. This is demonstrated by the 100% precision achieved by VF3. However, due to the function-preserving transformation, the explicit structure of $Q$ often disappears, leading to extremely low recall(0.32%) for VF3. These results highlight the importance of task definition.
>
>
>
>
> **Two-Stage Framework**
> Table 4 in Section 4.4 discusses the effectiveness of our proposed alignment, two-stage training and graph segmentation. The results demonstrate the effectiveness of each component in our method.
>
> Specifically, for fuzzy boundary identification, previous works on targeting arithmetic block identification (Subgraph Equivalence), such as Gamora [2] and ABGNN [3], focus on identifying exact input/output patterns. When identifying a half adder, Gamora, for instance, requires the root of the subgraph to be a XOR gate. However, after function-preserving transformations, such explicit boundaries may disappear (as discussed in Section 2.4), leading to significant performance degradation for these methods.
>
> Furthermore, considering that the graph encoder in our method can be replaced with other backbones, we evaluate our approach using alternative encoders designed for function-related tasks and propose several baselines, as shown in Tables 4 and 5 in Appendix B. The results demonstrate that with our proposed framework, these models gain significant performance improvement, indicating the importance of our proposed two-stage framework.
>
>
> **Function learning ability of GNN model**
> Prior works on subgraph isomorphism mainly focus on designing GNN models to learn structural patterns rather than functional behavior. As a result, these models perform poorly on function-related tasks, as discussed in prior studies [4,5].
>
> For function-related method, like Gamora and ABGNN, we integrate these graph encoders into our framework, and the results are as shown in Tables 4 and 5 in Appendix B. Even though with a significant performance improvement, these methods still show a significant performance drop compared to our method, indicating that the GNN architecture also leading to performance gap.
>
> *[1] Challenging the time complexity of exact subgraph isomorphism for huge and dense graphs with VF3 - Carletti V., Foggia P., Saggese A., Vento M. - IEEE transactions on pattern analysis and machine intelligence - 2018*
>
> *[2]Wu N, Li Y, Hao C, et al. Gamora: Graph learning based symbolic reasoning for large-scale boolean networks[C]//2023 60th ACM/IEEE Design Automation Conference (DAC). IEEE, 2023: 1-6.*
>
> *[3]Wang Z, He Z, Bai C, et al. Efficient arithmetic block identification with graph learning and network-flow[J]. IEEE Transactions on Computer-Aided Design of Integrated Circuits and Systems, 2022, 42(8): 2591-2603.*
>
> *[4]Liu J, Zhai J, Zhao M, et al. Polargate: Breaking the functionality representation bottleneck of and-inverter graph neural network[C]//Proceedings of the 43rd IEEE/ACM International Conference on Computer-Aided Design. 2024: 1-9.*
>
> *[5]Zheng Z, Huang S, Zhong J, et al. DeepGate4: Efficient and Effective Representation Learning for Circuit Design at Scale[C]//The Thirteenth International Conference on Learning Representations.*
>
>
> **Q2: Overall, I think the novelty of the proposed method is limited in some sense.... Overall, the key concept is to find the functional equivalence rather than structural isomorphism. More discussion about this concern is preferred.**
>
>
> A2: Thanks for the suggestion from reviewer. We would like to emphasize that the components of our proposed method stem from **the novel insight behind our proposed concept**, which has never been discussed in previous work. Our contribution can be summarized in three points.
>
>
> First, to the best of our knowledge, we for the first time introduce and formally define the novel concept of **functional subgraph matching**, clearly distinguishing it from structural isomorphism and functional equivalence. We also want to highlight that the definition itself plays a crucial role. As discussed in Q1, we evaluated the state-of-the-art heuristic for subgraph isomorphism, VF3. The comparison between VF3 and our method clearly highlights how differences in task formulation significantly impact the effectiveness of existing approaches.
>
> Second, we develop a **two-stage multi-modal** embedding framework, leveraging both intra-modal and inter-modal alignment to capture structure-agnostic and function-invariant graph representations. This allows effective functional subgraph detection across different modalities. While previous works focus on subgraph matching within single modality, we are the first work that explore cross-modal subgraph matching.
>
> We further propose alignment to acquire structure-agnostic and function-invariant embedding. The insight of alignment comes from the property of our proposed functional subgraph relation. Specifically, let's first go back to the definition of Functional Subgraph:
>
> **Definition(Functional Subgraph)**: A graph $\mathcal{Q}$ is a functional subgraph of $\mathcal{G}$, denoted $\mathcal{Q} \preccurlyeq \mathcal{G}$, if there exists a graph $\mathcal{G}'$ such that $\mathcal{G}' \equiv_{func} \mathcal{G}$ and $\mathcal{Q}$ is isomorphic to a subgraph of $\mathcal{G}'$.
>
> According to the definition, if $Q\preccurlyeq G_1$ and $G_1\equiv_{func}G_2$, then $Q\preccurlyeq G_2$. This denotes that the subgraph relation is preserved under functional equivalence **on the right-hand side** of the relation $\preccurlyeq$.
>
> In addition, we proposed a property, Functional Equivalence Preservation, in Section 2.3.
>
> **Proposition(Functional Equivalence Preservation)**: If $\mathcal{G}_1$ is a functional subgraph of $\mathcal{G}_2$, and $\mathcal{G}'_1$ is functionally equivalent to $\mathcal{G}_1$, then $\mathcal{G}'_1$ is a functional subgraph of $\mathcal{G}_2$.
>
> This property shows that the subgraph relation is preserved under functional equivalence **on the left-hand side** of the relation $\preccurlyeq$.
>
> These two results imply that, if the subgraph relation $Q\preccurlyeq G$ hold, then if we replace $Q$ or $G$ with another functional equivalent graph, the subgraph relation continues to hold. This invariance is the **key insight** for our alignment: the embeddings of functionally equivalent graphs should be aligned, regardless of their structural variations.
>
> Last, we propose an innovative approach for fuzzy boundary identification by formulating the task as a graph segmentation problem rather than a simple input-output classification problem, significantly enhancing boundary continuity and localization accuracy. This also comes from the insight that the clear boundary disappear after transformation.
>
> As discussed in Q1, previous works like Gamora and ABGNN, focus on identifying exact input/output patterns. When identifying a half adder, Gamora, for instance, requires the root of the subgraph to be a XOR gate. However, after function-preserving transformations, such explicit boundaries may disappear (as discussed in Section 2.4), leading to significant performance degradation for these methods.

---

> ### Comment · Area_Chair_29Qu · 2025-08-05
>
> Dear Reviewer,
>
> This is a kind reminder that the discussion phase will be ending soon on Aug 8th. Please read the author's responses and engage in a constructive discussion with the authors.
>
> Thank you for your time and cooperation.
>
> Best,
>
> NeurIPS 2025 Area Chair

---

### Note · Authors · 2025-08-12

We thank all reviewers for their thoughtful evaluations and constructive feedback. We are encouraged by the recognition of our work’s novelty, clear motivation, and strong empirical results. Several reviewers highlighted the originality of defining functional subgraph matching for the first time, moving beyond structural isomorphism and enabling robustness to synthesis and technology-mapping transformations.

We clarified that our contributions(**Reviewer apyh, Reviewer WZEu**) stem from the definition itself, which induces a fundamentally different task formulation. This motivated our two-stage multi-modal alignment framework and the formulation of fuzzy boundary identification as a segmentation problem. We are also the first to enable cross-modal subgraph matching. Ablations (Sec. 4.4) and alternative backbone experiments (Appendix B) show consistent gains across backbones, demonstrating improvements are model-agnostic.

On scalability(**Reviewer Ytof, Reviewer tTFV**), the main challenge is functional diversity in large designs and extreme size imbalance in small–large retrieval tasks, rather than computational complexity. We discussed practical strategies (e.g., divide-and-conquer) and provided medium-scale experiments showing state-of-the-art results despite distribution shifts, demonstrating feasibility for larger circuits.

Regarding runtime(**Reviewer Ytof, Reviewer tTFV**), our method is competitive among learning-based approaches while achieving orders-of-magnitude speedups over heuristics in functional settings. The minor overhead versus other learning-based methods is outweighed by substantial accuracy gains, and further reductions are possible via GNN optimizations, which are not the focus of our work.

For comparisons with heuristics(**Reviewer tTFV**), we explained that existing heuristics (e.g., VF3) or naive algorithms cannot directly address functional matching due to task definition and the exponential complexity of enumerating all functionally equivalent graphs. Empirically, VF3’s recall drops to 0.32% in functional settings despite perfect precision in isomorphism, underscoring the necessity of our formulation.

In summary, our work introduces a new problem, proposes a general and effective framework, and demonstrates substantial performance gains over both structural and function-related baselines. We believe these contributions justify acceptance and will inspire follow-up research in scalable, function-aware subgraph matching.

---

### Decision · Program_Chairs · 2025-09-17

**Decision:**

Accept (poster)

**Comment:**

The paper introduces an approach called functional subgraph matching to identify logic subgraphs in circuits based on functionality rather than structural isomorphism. All the reviewers agreed that the problem is well-motivated, the proposed approach is novel, and the empirical evaluations are thorough. Overall, this paper is well written and easy to follow, and the idea of functional subgraph matching is valuable for this community. The authors' rebuttal addressed most of the concerns of reviewers, and all the reviews tend to accept the paper during the reviewer-author discussion phase. Therefore, I recommend this paper to the NeurIPS 2025 conference.